# FlowState: Sampling-Rate-Equivariant Time-Series Forecasting

Lars Graf [1 2]   Thomas Ortner [1]   Stanisław Woźniak [1]   Angeliki Pantazi [1]

## Abstract

Existing time series foundation models (TSFMs), often based on transformer variants, lack adaptability to different sampling rates, struggle with generalization across varying context and target lengths, and are computationally inefficient. We introduce FlowState, a novel TSFM architecture that achieves sampling-rate-equivariant forecasting through a unified design that pairs a state space model (SSM) encoder with a functional basis decoder (FBD). This design enables continuous-time modeling and dynamic timescale adjustment, allowing FlowState to inherently generalize across all possible temporal resolutions, and dynamically adjust the forecasting horizons without retraining. We further propose an efficient pretraining strategy that improves robustness and accelerates training. Despite being one of the smallest TSFMs, FlowState achieves state-of-the-art results on the widely used GIFT-Eval benchmark, while demonstrating superior adaptability to unseen sampling rates. Our detailed analyses confirm the effectiveness of its components, and we demonstrate its unique ability to adapt to varying input sampling rates.

## 1. Introduction

Machine learning (ML) models are ubiquitously found in many aspects of our daily lives. Especially foundation models (FMs), have received extensive research interest and are today employed in various natural language processing (NLP) tasks, such as text summarization, text generation or information retrieval from large, unstructured text databases (Hadi et al., 2023). The astounding capabilities of FMs are believed to arise in part from their specific training process, in which the FMs are trained on a vast collection of text-based data available on the public internet. Through this approach, the model can learn the underlying foundational principles of the data. Thus, we refer to this training procedure as the foundation model approach. Leveraging this extracted information, the model can then address various unseen downstream tasks, i.e., is capable of zero-shot generalization (Bommasani et al., 2021).

Despite their astonishing performance in NLP, FMs struggle to be applied to other domains, such as time series processing. NLP and time series processing are both sequence processing tasks, but with major differences. In particular, a token in NLP carries substantially more information than an individual time series data point. Furthermore, time series data can be multivariate and vary strongly within and across domains, e.g., the electrical power consumption of a city may look entirely differently than a stock price.

Therefore, compared to the NLP domain, other model capabilities are required, which resulted in different model architectures to emerge as state-of-the-art (SOTA). For example, while FMs for NLP have become dominated by the transformer architecture, the same architecture is performing poorly in time series tasks (Zeng et al., 2022). Researchers uncovered better architectures based on linear layers that mix over time and features (Chen et al., 2023; Ekambaram et al., 2023) in an alternating manner. Recently, state space models (SSMs) (Gu et al., 2021) have emerged as viable alternatives with SOTA results in several time series tasks.

The above-mentioned challenges have for a long time hindered the emergence of time series foundation models (TSFMs). Only recently, researchers have found ways to utilize the foundation model training approach also for time series data and successfully trained TSFMs (Auer et al., 2025; Ansari et al., 2024; Ekambaram et al., 2024; Das et al., 2023; Wang et al., 2025a; Liang et al., 2024). Still, these approaches are quite limited in their usability. For example, a TSFM should be able to process time series data of a particular length—the context—and produce a forecast of a different and potentially varying length—the target. However, most TSFMs currently employ a linear decoder of a fixed size, matched to the target, to produce their forecast. This results in the models being inherently specialized to a specific target length. On top of that, TSFM should generalize well to varying context and target lengths, where current TSFMs often struggle. Finally, current TSFMs don't have a

---

[1]IBM Research Europe – Zurich, Switzerland [2]University of Zurich and ETH Zurich, Switzerland. Correspondence to: Lars Graf <Lars.Graf1@ibm.com>.

*Proceedings of the 43rd International Conference on Machine Learning*, Seoul, South Korea. PMLR 306, 2026. Copyright 2026 by the author(s).

direct way to adjust during evaluation to the specific characteristics of the time series, such as a change in sampling rate.

In this work, we propose a novel TSFM, called FlowState[1], that addresses these shortcomings. In particular, FlowState is based on an SSM, which naturally allows to process varying context lengths. Moreover, we introduce a novel functional basis decoder (FBD) which leverages a set of basis functions to create a continuous forecast. Most importantly, FlowState is entirely designed to be sampling-rate equivariant. The SSM encoder translates the inputs into sampling-rate invariant hidden states, while the FBD interprets them as coefficients of a functional basis, producing a continuous forecast, sampleable at any rate. FBD also enables the model to produce varying target lengths, depending on the time-scale of the data. Finally, we designed a massively parallel training scheme, that concurrently trains the model on a variety of context lengths for superior generalization.

In summary, we make the following key contributions:

- FlowState: We present a sampling-rate equivariant SSM-based time series foundation model that can be dynamically adjusted to the specific characteristics of the time series

- Functional basis decoder (FBD): We propose a novel decoder, as a critical component of FlowState, that utilizes a set of continuous basis functions to allow seamless adjustment to specific input characteristics and to produce forecasts for varying target lengths

- Training approach with parallel predictions: We introduce a foundation model training scheme leveraging parallel predictions to enable efficient training and model robustness to varying context lengths

## 2. Background

### 2.1. Time Series Forecasting

Time series data is omnipresent in various domains and there are several tasks that can be performed with this data, such as anomaly detection, forecasting, pattern search within time series, etc. Although our model could be applied to several of these tasks, we specifically focus on time series forecasting. The model receives an input time series $\mathbf{X} \in \mathbb{R}^{L \times c} = \{\mathbf{x}_1, ..., \mathbf{x}_L\} = \mathbf{x}_{1:L}$, where $\mathbf{x}_t \in \mathbb{R}^c$ is the $c$-channel multivariate time series at timestep $t$ and $L$ is the context length. Given this input data, the task is to produce a forecast for the proceeding $T$ timesteps, i.e., to produce $\hat{\mathbf{Y}} \in \mathbb{R}^{T \times c} = \{\hat{\mathbf{y}}_1, ..., \hat{\mathbf{y}}_T\} = \hat{\mathbf{y}}_{1:T} = \mathbf{x}_{L+1:L+T}$, where $T$ is the forecasting length. The quality of the forecast

can be measured by comparing it against the ground truth $\mathbf{Y} \in \mathbb{R}^{T \times c}$ using various metrics, such as the mean absolute error (MAE) or the mean squared error (MSE).

### 2.2. Models for time series data

Traditionally time series forecasting tasks have been addressed with classic machine learning models, such as the ARIMA model (Box et al., 2015), which to this day presents a strong baseline in some cases.

The successes of the transformer architecture in the NLP domain inspired researchers to apply them to time series data (Das et al., 2024). Despite high performance, these models are typically large and require large datasets and training tricks. Vanilla transformers were outperformed by simpler architectures (Zeng et al., 2022). Combining several timesteps to patches (Nie et al., 2023), or alternating self-attention along the time and the feature dimension (Liu et al., 2024b), improved their performance over classic baselines for some datasets. However, they still struggle on certain datasets and suffer from a high computational cost. This has inspired researchers to develop simpler approaches, solely based on multi-layer perceptrons (MLPs), that outperform transformers with a smaller computational footprint (Chen et al., 2023; Ekambaram et al., 2023).

Recently, State Space Models have emerged as another viable alternative. SSMs have a long history in control theory, are successfully applied to NLP tasks, and slowly make their way into other domains as well (Liu et al., 2024a; Rahman et al., 2024; Wang et al., 2025b). In contrast to the transformer- and MLP-based architectures, SSMs are stateful models more similar to the classic recurrent neural networks (RNNs), which in the past also served as strong baselines in time series tasks (Siami-Namini et al., 2019; Che et al., 2018). Researchers have developed several SSM variants for NLP tasks, such as S4 (Gu et al., 2022b), S5 (Smith et al., 2023), S6 (Gu & Dao, 2023) or Mamba2 (Dao & Gu, 2024), which successively enhanced capabilities and performance. A main advantage of SSMs over RNNs is that while the state update in conventional RNNs, such as LSTMs or GRUs, contains nonlinearities, the state update of SSMs is linear, and only the output of the SSM contains a nonlinear activation. Thus, SSMs can be parallelized, which leads to a reduced runtime compared to sequential RNNs.

Lastly, TiRex (Auer et al., 2025), a stateful model based on xLSTMs (Beck et al., 2024), has emerged and became state-of-the-art in several benchmarks. In addition, it introduced a capability to deal with various forecasting lengths by applying a novel Multi-Patch-Inference (MPI) process.

Typically TSFMs use autoregressive techniques to extend their forecasting horizon. Namely, they produce several shorter forecasts of size $p < T$ sequentially, appending their

---

[1] https://huggingface.co/ibm-research/flowstate

forecast to the original context. Auer et al. (2025) have improved this autoregressive technique with MPI. In particular, they adopt contiguous patch masking (CPM) during training, which accustoms the model to make a prediction after a certain number of unknown timesteps. This setup enables MPI to forecast future patches by treating intermediate ones as missing. The main advantage of MPI / CPM over autoregressive forecasting is the increased model's robustness to noise and uncertain data, as well as the ability to propagate uncertainty over multiple forecasting patches.

## 3. FlowState

Our proposed model, FlowState, is an encoder-decoder architecture, employing an S5-based encoder and a functional basis decoder. Figure 1a shows an overview of its architecture. The input time series with length $L$ is first normalized in a causal manner. This causality is critical, because of the parallel forecasts performed during training, see Section 5. Afterwards, the normalized inputs are embedded linearly and provided to the SSM encoder directly without any patching, see Section 3.1 for details. Importantly, while the time series before being processed by the SSM are considered to be in the feature space, where each input element represents features of the time series, the SSM encodes this information into a coefficient space, where it operates on coefficients of continuous basis functions. The final output of the SSM encoder forms the basis for the FBD, see Section 3.2 for details, whose outputs are then inverse normalized, using the inverse method of the input normalization, and form the forecasts of our model. Importantly, the FBD maps from the coefficient space back to the feature space to provide the forecasts. Furthermore, the SSM encoder, as well as the FBD are controlled by additional scaling factors $s_{\Delta_E}$ and $s_{\Delta_F}$, respectively. These factors allow to adjust the SSM encoder and the FBD to the sampling rate of the input data, making FlowState sampling-rate equivariant.

### 3.1. SSM Encoder

The SSM encoder is a stack of SSMs layers, each consisting of an SSM block followed by an MLP, see Figure 1b. The SSM block is inspired by the S5, with an output gate for improved selectivity. Its dynamics can be described as:

$$\mathbf{s}_t^l = \bar{\mathbf{A}}^l \mathbf{s}_{t-1}^l + \bar{\mathbf{B}}^l \boldsymbol{x}_t^{l-1} \tag{1}$$

$$\mathbf{c}_t^l = \bar{\mathbf{C}}^l \mathbf{s}_t^l \tag{2}$$

$$\mathbf{h}_t^l = \mathbf{c}_t^l \cdot \text{gate}_o^l(\mathbf{c}_t^l) + \bar{\mathbf{D}}^l \boldsymbol{x}_t^{l-1} = \text{SSM}_{\Delta_E^l}(\boldsymbol{x}_t^{l-1}) \tag{3}$$

where $\text{gate}^l(\mathbf{c}_t^l) = \sigma(\mathbf{W}^l \mathbf{c}_t^l + \mathbf{b}^l)$ is the output gate for the SSM, $\bar{\mathbf{A}}^l \in \mathbb{R}^{P \times P}$, $\bar{\mathbf{B}}^l \in \mathbb{R}^{c \times P}$, $\bar{\mathbf{C}}^l \in \mathbb{R}^{P \times H}$ and $\bar{\mathbf{D}}^l \in \mathbb{R}^{c \times H}$ are the state transition, the input, the output and the skip connections matrices of layer $l$, $m$ and $n$ are

the hidden state size and the output size of the SSM block and $\mathbf{s}_t^l$ and $\mathbf{h}_t^l$ are the state and the output of the SSM block at timestep $t$. The input is denoted as $\boldsymbol{x}_t^0$. As reported in (Smith et al., 2023), the discrete matrices of the SSM block $l$ can be computed from the continuous ones by using the zero-order-hold (ZOH) method:

$$\bar{\mathbf{A}}^l = e^{\mathbf{A}^l \Delta_E^l}, \bar{\mathbf{B}}^l = \mathbf{A}^{l-1}\left(\bar{\mathbf{A}}^l - 1\right)\mathbf{B}^l, \bar{\mathbf{C}}^l = \mathbf{C}^l, \bar{\mathbf{D}}^l = \mathbf{D}^l,$$

where $\text{diag}(\mathbf{A}^l) \in \mathbb{R}^P$, $\mathbf{B}^l \in \mathbb{R}^{c \times P}$, $\mathbf{C}^l \in \mathbb{R}^{P \times H}$, $\mathbf{D}^l \in \mathbb{R}^{c \times H}$ and $\Delta_E^l \in \mathbb{R}^P$ are the trainable parameters of block $l$, initialized using the HiPPO method (Gu et al., 2023). Subsequently, the output of each SSM block is further processed by an MLP layer, see Figure 1b.

Note, in contrast to several other SOTA works (Auer et al., 2025; Ekambaram et al., 2024; Ansari et al., 2024; Cohen et al., 2024; Das et al., 2024), FlowState processes data as-is and doesn't perform any quantization or patching. The SSM architecture can naturally be adjusted to a change of the input sampling rate, as also demonstrated in Smith et al. (2023). By adjusting $\Delta_E$, the SSM can produce similar representations, irrespective of the sampling rate. While this effect can be beneficial for classification and regression tasks (Smith et al., 2023), it is insufficient for time series forecasting tasks. In particular, current decoders cannot distinguish those similar representations, hence they cannot adjust the forecasting sampling rate properly. To remedy this issue, we propose a novel decoder that builds on top of the SSM encoder.

### 3.2. Functional Basis Decoder

For the functional basis decoder, we take inspiration from how SSMs are initialized from an input sequence. The HiPPO approach ensures that their hidden state expresses coefficients of a polynomial basis, which optimally approximates the input sequence. In particular, Gu et al. (2020) demonstrated a possibility to use the hidden state of their SSM at timestep $t$ to reconstruct the input sequence until $t$ with a functional basis. We adopt this approach for our decoder, but instead of extracting coefficients that can be used to reconstruct the input, we use a continuous functional basis to construct the forecast from the final outputs of the SSM encoder $\mathbf{o}_L^N$, see Figure 1c. In particular, our proposed FBD interprets the final outputs of the SSM encoder, $\mathbf{o}_L^N$, as coefficients of a functional basis, which can in turn be used to produce a continuous output function. To obtain the forecast with a desired sampling rate, this continuous output is then sampled at an equally spaced interval, with the spacing

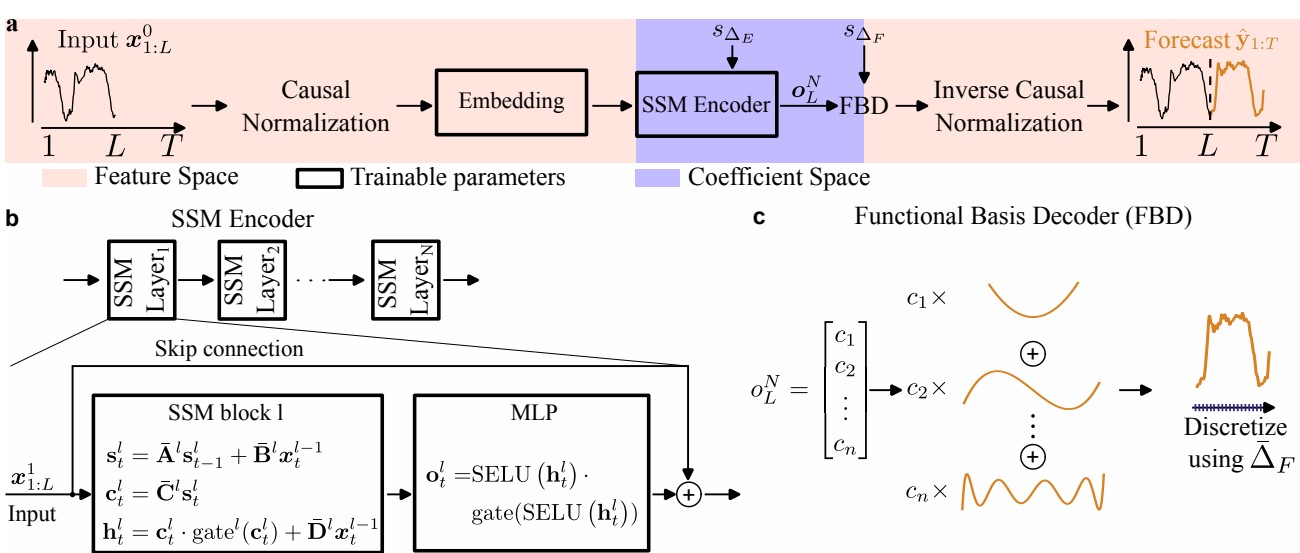

*Figure 1.* **Architecture overview. a** Overview of the architecture. The input context in the feature space (orange color) is normalized, embedded and then processed by the SSM encoder. The SSM encoder transforms the input into the coefficient space (blue color) and provides the final encodings to the functional basis decoder, which then produces the forecast. Modules with trainable parameters are highlighted in black rectangular blocks. **b** The SSM encoder comprises $N$ S5 layers, each composed of an S5 block extended with an MLP layer. A skip connection is used to propagate inputs to later encoder layers. **c** The functional basis decoder interprets the outputs $\mathbf{o}_t^l$ of the SSM encoder as coefficients of a functional basis and creates a continuous output, which is sampled to produce the final forecast.

$\Delta_F$. Thus, the FBD can be formalized as follows:

$$c_i = o_{L,i}^N \tag{4}$$

$$\tilde{\mathbf{y}} = \sum_{i=1}^n c_i p_i(a,b) \tag{5}$$

$$\hat{\mathbf{y}} = \text{sample}(\tilde{\mathbf{y}}, \Delta_F) = \text{FBD}_{\Delta_F}(\mathbf{o}_L^N) \tag{6}$$

where $p_i(\cdot,\cdot)$ is the $i$-th basis function evaluated at an interval $[a,b]$, $\tilde{\mathbf{y}}$ is the continuous forecast and sample$(\cdot, \Delta_F)$ samples the argument equally spaced with $\Delta_F$.

Our functional basis decoder offers several key advantages. Firstly, it produces a continuous forecast, which can then be sampled with any desired sampling rate. Secondly, it draws inspiration from a well-established procedure to map from coefficient to feature space, and thus can leverage various functional basis functions, depending on the task. For our main experiments, we use the Legendre polynomials to be consistent with the SSM input encoding used by the HiPPO initialization. Another viable option is to use the Fourier basis functions to better match periodic signals. Finally, and most importantly, it enables the decoder to produce forecasts at the desired sampling rate, see next section. Note that although we introduce the FBD as part of FlowState, it is a separate component and can be combined with other encoder architectures as well.

### 3.3. Adjusting $\Delta$ for unseen sampling rates

As described above, the dynamics of the SSM encoder and the FBD can be adjusted to the input sampling rate by modi-

fying $\Delta_E$ and $\Delta_F$. In conventional SSMs, these parameters are adjusted only during training, while during inference they remain fixed. In order to enable adjustments to the sampling rates during inference, we modify them with an additional scaling parameter, in particular:

$$\bar{\Delta}_{\{E,F\}} = f\left(\Delta_{\{E,F\}}, s_{\Delta,\{E,F\}}\right) = s_{\Delta,\{E,F\}} \cdot \Delta. \tag{7}$$

This adaptation of the parameter $\Delta_{\{E,F\}}$ by multiplication with the scale factor $s_{\Delta_{\{E,F\}}}$ is a central novelty of our FBD and crucial to discretize the continuous SSM for a given sampling rate.

## 4. Sampling-Rate Equivariance

FlowState is designed to be *sampling-rate equivariant*. In particular, when $s_{\Delta_E} = s_{\Delta_F}$, a change of the sampling rate of the context leads to the same change of the sampling rate of the target, without affecting FlowState's underlying continuous-time forecasting. Moreover, when $s_{\Delta_E} \neq s_{\Delta_F}$, it can produce forecasts at sampling rates different from those of its input.

To illustrate this property, let $\mathcal{F}_\Delta(\mathbf{x}) = \text{FBD}_\Delta(\text{SSM}_\Delta(\mathbf{x}))$ denote the end-to-end FlowState predictor operating on $\Delta$-sampled inputs and returning predictions on the same $\Delta$-grid. For simplicity we assume that the FlowState Encoder consists only of a single SSM. For any $\Delta, \Delta' > 0$, we show

$$\mathcal{F}_{\Delta'}(\mathbf{x}) \approx \text{FBD}_{\Delta'}(\text{SSM}_\Delta(\mathbf{x})). \tag{8}$$

In other words, sampling the continuous input $\mathbf{x}$ at a different rate yields (approximately) the same latent state and thus

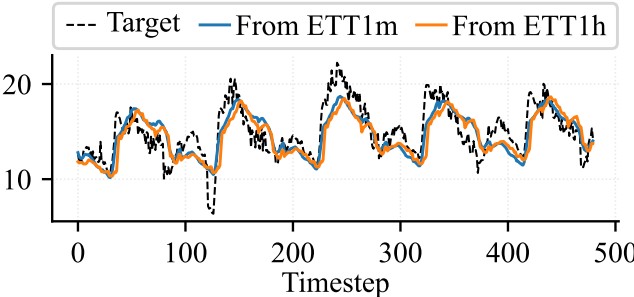

*Figure 2.* **Demonstration of sampling-rate equivariance**. The black line represents the ground-truth target for a sample from the ETT1m dataset. The blue line shows the prediction of FlowState in the common way, i.e., providing ETT1m data as context and setting $s_{\Delta_E} = s_{\Delta_F} = 0.25$. For the orange line we provide ETT1h data as context to FlowState and set $s_{\Delta_E} = 1.0$ and $s_{\Delta_F} = 0.25$. This way, we produce ETT1m predictions from a ETT1h context, showcasing the sampling-rate equivariance of FlowState.

the same continuous forecast, which can then be evaluated on any grid.

**Encoder: approximate invariance across step sizes.** The encoder implements a fixed continuous-time SSM, see Section 3.1, discretized by a ZOH mechanism with step size $\Delta$ (Åström & Wittenmark, 2013). The ZOH discretization method assumes a piecewise constant input, which for a general continuous input $\boldsymbol{x}(t)$ introduces a global discretization error $e_{\text{ZOH},\Delta}$ at the last timestep T, which approaches zero as the step size $\Delta$ approaches zero.

Consequently, discretizing the *same* SSM at step sizes $\Delta$ and $\Delta'$, resulting in final states $s_T, s'_T$, yields the stability bound

$$\|s_T - s'_T\| \leq e_{\text{ZOH},\Delta} + e_{\text{ZOH},\Delta'} \qquad (9)$$

and hence the encoder representation is approximately invariant to the sampling rate and bound by the approximation error determined by the ZOH approximation in the SSM.

$$\text{SSM}_\Delta(\boldsymbol{x}) \approx \text{SSM}_{\Delta'}(\boldsymbol{x}).$$

**Decoder: rate-agnostic continuous forecasting** The FBD first maps the encoder output $\text{SSM}_\Delta(\boldsymbol{x})$ to a continuous forecast $\tilde{\mathbf{y}}(t)$, see Section 3.2, which is agnostic to sampling rates. The sampling step $\Delta$ enters only in the final prediction step, i.e., $\hat{\mathbf{y}} = \text{sample}(\tilde{\mathbf{y}}, \Delta_F)$. Thus, changing $\Delta$ only changes the output grid.

**Combined sampling-rate equivariance.** Using Equation 8 we can define the equivariance error as,

$$e_{\text{equi}} := \mathcal{F}_{\Delta'}(\boldsymbol{x}) - \text{FBD}_{\Delta'}(\text{SSM}_\Delta(\boldsymbol{x})) \qquad (10)$$
$$= \text{FBD}_{\Delta'}(\text{SSM}_{\Delta'}(\boldsymbol{x})) - \text{FBD}_{\Delta'}(\text{SSM}_\Delta(\boldsymbol{x})) \qquad (11)$$
$$\leq O(e_{\text{ZOH},\Delta} + e_{\text{ZOH},\Delta'}) \qquad (12)$$

where in the last equation we used Equation 9 as well as the fact that the FBD is rate agnostic and performs the sampling on the time-continuous forecast. Therefore, FlowState is approximately sample-rate equivariant, with an approximation error vanishing as $\max\{\Delta, \Delta'\} \to 0$.

Figure 2 empirically demonstrates this equivariance. In this experiment we aim to forecast data from the ETT1m dataset, which is sampled with a $\Delta' = 15$-minute interval, see Appendix C for more details on the dataset. We plot the predictions of FlowState ($\text{FBD}_{\Delta'}(\text{SSM}_{\Delta'}(\boldsymbol{x}))$) in blue. In addition, we provide the context of the ETT1h data, which was sampled with a $\Delta = 1$-hour interval, and plot the predictions of $\text{FBD}_{\Delta'}(\text{SSM}_\Delta(\boldsymbol{x}))$ in orange. In other words, the orange line showcases predictions for ETT1m targets from a ETT1h context. We also examine the invariance of the SSM encodings as well as a the formal analysis in more detail in the Appendix B.

## 5. Parallel forecasts

To enable efficient training of FlowState and to enhance its robustness to varying context lengths, we introduce an advanced foundation training scheme. In particular, it utilizes multiple parallel forecasts with increasingly longer contexts, ranging from $L_{\min}$ to $L$, see Figure 3. This training technique only affects the pretraining of our model, but neither the architecture itself nor the inference process.

These various forecasts can be formulated as

$$\hat{\boldsymbol{y}}_{t+1:t+T} = \text{FBD}_{\Delta_F}\left(\text{SSM}_{\Delta_E}\left(\boldsymbol{x}^0_{1:t}\right)\right), \qquad (13)$$

where $t \in [L_{\min}, L]$. Importantly, because FlowState is an SSM-based architecture, the inputs can be processed in parallel and in turn also the multiple forecasts can be produced in parallel. Thus, this approach allows to produce $(L - L_{\min})$ forecasts from any given context-target pair, while other models traditionally only produce a single forecast per context-target pair. Depending on the choice of $L$ and $L_{\min}$, this amount can be very large, for example for $L = 4096$ and $L_{\min} = 20$, as was used for our main results, FlowState produces 4076 forecasts per sample in parallel.

The benefits of this training scheme can either materialize in significantly reduced training times, because one can iterate through the dataset faster using a larger stride to the next context-target pair, or in an increased number of training examples, when the original stride to the next context-target pair is kept constant. Another advantage of this training procedure is that the model inherently learns to produce forecasts from varying context lengths. This naturally increases the models' generalization capabilities and robustness. Note that our novel training scheme is not limited to the FlowState architecture but can be applied to any causal architecture that can produce multiple forecasts in parallel.

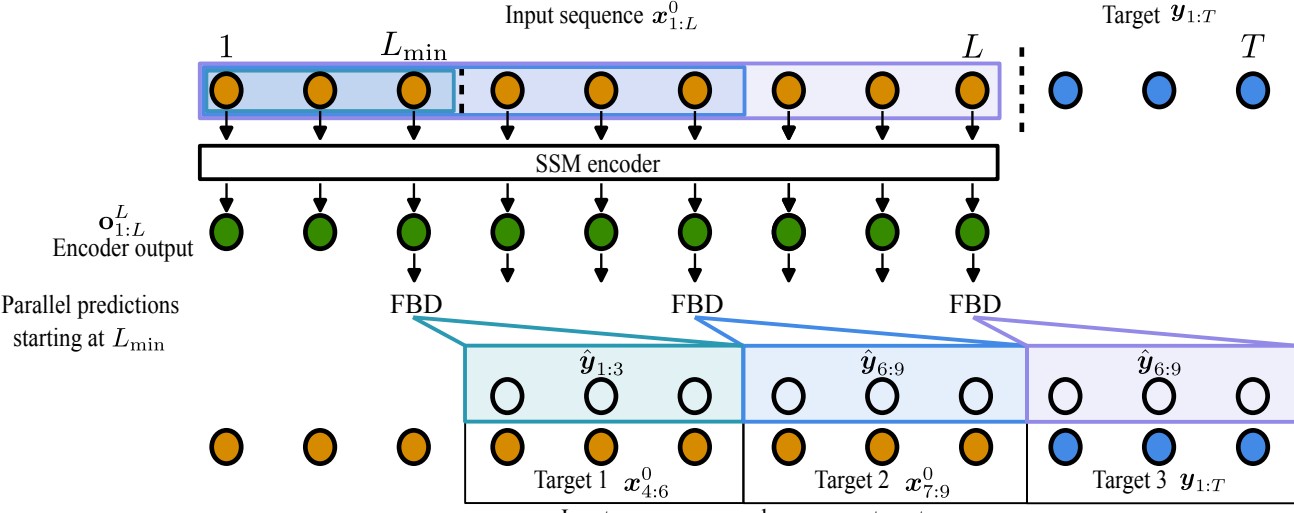

*Figure 3.* **Schematic illustration of our parallel prediction training scheme.** The input sequence $\mathbf{x}^0_{1:T}$ is encoded in parallel using the SSM encoder. Starting from $L_{\min}$, multiple forecasts are produced in parallel, where each forecast has its own target and is based on an increasing context length. In particular, a prediction is made for every timestep after $L_{\min}$, but for clarity only three are shown. For example, for the first forecast (green color) only the first three timesteps $\mathbf{x}^0_{1,2,3}$ are used as the context, while for the last prediction (purple color) the full context $\mathbf{x}^0_{1:T}$, is used. Note that for some of the forecasts the input sequence itself serves as the target and thus a causal processing of the input is essential to avoid information leakage.

Note, parallel forecasting can also speed-up inference if combined with MPI.

### 5.1. Causal normalization

For parallel forecasting to work properly an architecture has to be strictly causal. Otherwise, the model may learn to exploit this information leakage during training. In particular, the prediction $\hat{\mathbf{y}}_{t+1:t+T}$ should only use the data from $\mathbf{x}^0_{\leq t}$. The SSM and FBD of FlowState naturally satisfy this requirement, but the commonly applied normalization and inverse normalization technique, RevIN (Kim et al., 2022), would violate it.

RevIN normalizes every context-target pair, based on the mean and the standard deviation of the entire context $\mathbf{x}^0_{1:L}$, see Eq. 2 of (Kim et al., 2022). However, this would result in information from $\mathbf{x}^0_{>t}$ to influence $\hat{\mathbf{y}}_{t+1:t+T}$. For example, if the average of the normalized sequence $\mu_t = \tilde{\mathbf{x}}^0_{1:t}$ is negative, the model will learn that positive values are to be expected for $\tilde{\mathbf{x}}^0_{>t}$, because, per definition, RevIN produces a zero mean for the whole time series.

To address this problem, we use a causal form of RevIN. Specifically, instead of using the average and standard deviation of the entire context to normalize, we leverage a running mean and a running standard deviation. Each element of the input at time $t$ is then normalized using these

quantities at time $t$. This can be formulated as

$$\mu_{r,t} = \frac{1}{t} \sum_{i=1}^{t} x^0_i \tag{14}$$

$$\sigma^2_{r,t} = \frac{1}{t} \sum_{i=1}^{t} \left(x^0_i - \mu_{r,i}\right)^2 \tag{15}$$

$$\tilde{\mathbf{x}}^0_{1:t} = \frac{\mathbf{x}^0_{1:t} - \boldsymbol{\mu}_{r,1:t}}{\boldsymbol{\sigma}_{r,1:t}}, \tag{16}$$

where $\boldsymbol{\mu}_{r,1:t}, \boldsymbol{\sigma}_{r,1:t}$ are causal mean and the causal variance of the timesteps 1 to $t$.

Similarly, each forecast of the FBD is de-normalized by the statistics of the last timestep of the context. For example, $\hat{\mathbf{y}}_{t:t+T}$ is de-normalized with $\mu_{r,t}$ and $\sigma_{r,t}$.

## 6. Experiments

We pretrain the proposed FlowState model and then evaluate its forecasting capabilities on the competitive **GIFT-Eval** benchmark[2] (Aksu et al., 2024). For pretraining we use datasets from the GIFT-Eval-Pretrain corpus and from the Chronos Pretraining data. In addition, we added synthetic time series generated via CauKer (Xie et al., 2025). All data —real and synthetic— are further enhanced using augmentation techniques introduced in Auer et al. (2025). For all our models we use CPM during pretraining, which al-

---

[2] https://huggingface.co/spaces/Salesforce/GIFT-Eval

lows for MPI, as introduced in the Background section to construct longer forecasts, exceeding the FBD horizon. Additional details about the pretraining data are provided in the Appendix C.

## 6.1. Evaluation Setup

We assess the forecasting performance of FlowState in a zero-shot setting, following the standard evaluation protocols of the GIFT-Eval benchmark. In particular, we compare our results to the state-of-the-art zero-shot models from the GIFT-Eval leaderboard who offer code for verification, namely TimesFM-2.5 (Das et al., 2024), TiRex (Auer et al., 2025), Chronos-2-Synth (Ansari et al., 2025), Kairos (Feng et al., 2025), Sundial (Liu et al., 2025) and Toto (Cohen et al., 2024). Additional results and comparisons on standard time series benchmark tasks, such as ETTm1, ETTh1, Traffic, etc. are provided in Appendix A.1.

### 6.1.1. METRICS

The Gift-Eval benchmark uses two metrics for comparison, the Mean Absolute Scaled Error (**MASE**) for point forecasting accuracy and the quantile-based **CRPS** metric for the probabilistic forecasting accuracy. Before reporting, both metrics are normalized per task using a Seasonal Naive baseline. Final scores are reported as the geometric mean across all 97 tasks of the GIFT-Eval benchmark.

### 6.1.2. TEMPORAL SCALING

FlowState's continuous-time formulation introduces two key considerations during evaluation. First, we determine suitable *scale factors* $s_{\Delta_E}$ and $s_{\Delta_F}$ for each dataset. Since in GIFT-Eval the task is to produce the forecast with the same sampling rate as the context, we set $s_{\Delta_E} = s_{\Delta_F} = s_\Delta$. However, the datasets may vary in both sampling rates and temporal dynamics, thus we base the scale factors on *seasonality* rather than raw sampling frequency. For example, hourly temperature data typically exhibits a 24-step daily cycle, while weekly peak temperatures follow a seasonal pattern of $365/7 \approx 52$ steps. Even though a week contains 168 hours, a more appropriate scale factor between these two examples is determined by the ratio of their relative seasonality. We define a base seasonality of 24 and compute the scale factor as:

$$s_\Delta = \frac{\text{Base Seasonality}}{\text{Seasonality}}$$

This ensures that all datasets are mapped to a common temporal scale in the model's continuous space. A scale factor of 1 corresponds to seasonality 24. Additional implementation details are provided in Appendix D.

### 6.1.3. CONTEXT AND FORECASTING LENGTH

FlowState's architecture enables flexible adaptation of both context and forecasting lengths across datasets with diverse temporal resolutions. Unlike discrete models, where these lengths are fixed, FlowState operates in a scale-adjusted latent space, representing continuous signals. To maintain consistency with pretraining, we define effective lengths which the model actually uses, relative to the scale factor $s_\Delta$, corresponding to the pretraining context length of 4096 steps, and to the base forecasting length $T = 6$ (=a quarter of a season). Depending on $s_\Delta$ these effective lengths will change. In particular, the effective context length $L_{\text{eff}}$ and forecasting length $T_{\text{eff}}$ are computed as:

$$L_{\text{eff}} = \frac{L}{s_\Delta}, \quad T_{\text{eff}} = \frac{T}{s_\Delta}$$

This formulation is particularly beneficial for datasets with large seasonality, which typically require longer historical context and benefit from extended forecasting horizons. As seasonality increases, $s_\Delta$ decreases, resulting in larger $L_{\text{eff}}$ and $T_{\text{eff}}$. This allows FlowState to forecast far into the future for such datasets—precisely where long-range predictions are often most valuable.

## 6.2. Results

We evaluate three FlowState variants on GIFT-Eval: FlowState-3M, FlowState-10.6M, and FlowState-18.6M. For the main comparison, all variants use a 4k context length. FlowState-18.6M uses a larger MLP while keeping the rest of the architecture the same as for FlowState-10.6M.

Table 1 presents the normalized MASE and CRPS metrics on GIFT-Eval, alongside the strongest public zero-shot models ranked by ascending MASE on the full GIFT-Eval benchmark. FlowState achieves state-of-the-art performance: FlowState-18.6M obtains the best MASE and CRPS, while

| Model | #Params | MASE↓ | CRPS↓ |
|---|---|---|---|
| *FlowState-18.6M* | 18.6M | **0.701** | **0.487** |
| *FlowState-10.6M* | 10.6M | 0.704 | 0.490 |
| TimesFM-2.5 | 200M | 0.705 | 0.490 |
| *FlowState-3M* | **3M** | 0.712 | 0.496 |
| TiRex | 35M | 0.716 | 0.488 |
| Chronos-2-Synth | 120M | 0.720 | 0.496 |
| Kairos-50M | 50M | 0.742 | 0.548 |
| Kairos-23M | 23M | 0.748 | 0.554 |
| Sundial-Base | 128M | 0.750 | 0.559 |
| Toto-Open-Base-1.0 | 151M | 0.750 | 0.517 |

*Table 1.* GIFT-Eval results, sorted by ascending MASE. All Flow-State variants use a 4k context length.

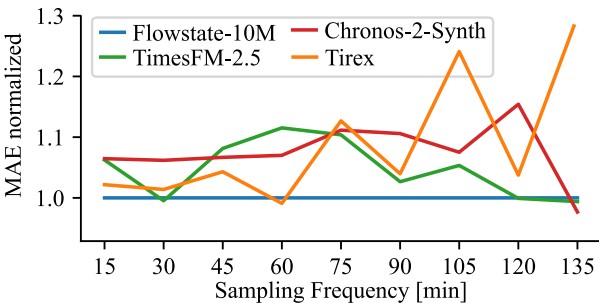

*Figure 4.* Normalized MAE performance across various sampling frequencies on the ETT1 dataset.

FlowState-10.6M outperforms all non-FlowState baselines in MASE. Compared to TimesFM-2.5, the strongest previous baseline by MASE, FlowState achieves better accuracy while using substantially fewer parameters.

The strong performance of the compact FlowState-3M and FlowState-10.6M variants demonstrates the parameter efficiency of the architecture. This suggests that sampling-rate equivariance reduces the need to separately learn patterns at each temporal resolution: patterns can be learned in a normalized sampling-rate space, while generalization across sampling rates follows from the equivariant architecture.

### 6.2.1. ROBUSTNESS TO UNSEEN SAMPLING RATES

To assess the robustness of FlowState to varying temporal resolutions, we conduct a controlled experiment on the ETT1m dataset. Originally sampled at 15-minute intervals, we sample the data to create versions with sampling intervals ranging from 15 to 135 minutes in 15-minute increments. We then evaluate FlowState-10.6M, TiRex, TimesFM-2.5 and Chronos-2-Synth on each version using the standard GIFT-Eval evaluation framework and a target length of 480 timesteps.

Figure 4 shows the MAE of all models for each sampling frequency, normalized vs. FlowState as baseline. FlowState-10.6M consistently outperforms all baselines across most frequencies, with a particularly large margin at uncommon sampling intervals. Some baseline models only bridge the performance gap at common frequencies, such as 15 min, 30 min, 60 min, etc., which were likely seen during pretraining. Moreover, while FlowState is able to naturally generalize to unseen sampling rates without retraining, other state-of-the-art models require exposure to every possible frequency during training.

### 6.2.2. ABLATION STUDY

To understand the contributions of individual components in FlowState, we conduct a series of ablations using the FlowState-3M model trained on 2k context, as opposed to

| Model Variant | MASE ↓ | CRPS ↓ |
|---|---|---|
| **FlowState-3M (2k)** | **0.725** | **0.502** |
| ± 3 seeds | $\pm 7 \times 10^{-4}$ | $\pm 11 \times 10^{-4}$ |
| *Core Components* | | |
| w/o equivariance | 0.799 | 0.553 |
| w/o parallel forecasts | 0.774 | 0.548 |
| *Encoder Ablations* | | |
| w/o output gate | 0.726 | 0.505 |
| S5 real | 0.862 | 0.602 |
| selective $\Delta_E$ | 0.749 | 0.521 |
| *Decoder Ablations* | | |
| fixed linear decoder | 0.754 | 0.526 |
| Fourier basis | 0.727 | 0.507 |
| full-Legendre basis | 0.730 | 0.507 |
| *Other Training Ablations* | | |
| w/o time noise | 0.740 | 0.518 |
| w/o causal RevIN | 0.738 | 0.513 |
| *Evaluation Variants* | | |
| 128 basis functions | 0.726 | 0.503 |
| 64 basis functions | 0.735 | 0.509 |

*Table 2.* **Ablation results** on FlowState-3M (2k context).

4k in the main results. We trained FlowState-3M (2k) three times using different seeds to get an estimate of the standard deviation. The mean and std of the baseline together with the ablations are summarized in Table 2, and organized into the following categories:

**Core Components.** This group isolates the impact of FlowState's key design choices. Removing the sampling-rate adjustment mechanism leads to a significant drop in performance, confirming the importance of sampling-rate equivariance. Disabling parallel predictions, by always only predicting from the last context point, also significantly degrades performance, though to a lesser extent.

**Encoder Ablations.** Removing FlowState's output gate, as introduced in Section 3.1, only results in a small decrease in MASE, and a more significant decrease in CRPS. Changing FlowState from a complex SSM to a real one (with initialization from S4D-real (Gu et al., 2022a)) leads to the strongest performance decrease. This indicates that without the complex domain the recurrent dynamics of S5 becomes insufficient for time series forecasting at variable sequence length.

One of the main improvements of S6 (Mamba) over its predecessors is selectivity: making B, C and $\Delta_E$ a function of the input. We've implemented selectivity on $\Delta_E$ similarly to S6, which affects the discrete SSM parameters $\bar{A}, \bar{B}$. We found that for FlowState making $\Delta_E$ selective clearly

decreases performance. This could indicate that FlowState's explicit adjustment of $\Delta_E$ with $s_\Delta$ does not pair well with selectivity on the same parameter.

**Decoder Ablations.** To isolate the contribution of the functional basis decoder (FBD), we replace it with a resolution-agnostic linear decoder operating on the same SSM encoder representations. This keeps the encoder unchanged while removing continuous decoding. Performance degrades from 0.725 to 0.754 MASE, indicating that the FBD is a key contributor. We attribute this drop to the loss of sampling-rate equivariance: while the encoder produces approximately invariant representations, the fixed decoder cannot generate a continuous forecast adaptable to different output resolutions, leading to local errors within prediction patches. Due to the use of Multi-Patch Inference, these errors are mainly local, while the large-scale prediction over longer horizons remains intact, explaining why the degradations are not more severe. Qualitative examples are provided in Appendix A.2.

Additionally, we evaluate two alternative sets of basis functions to the default Legendre basis: a Fourier basis, and "Full-Legendre", a Legendre basis trained to forecast four times longer horizons (an entire seasonality instead of only a quarter of a seasonality as in our main results). Both ablations resulted in a small but significant decrease in performance, both in MASE and CRPS.

**Other Training Ablations.** We further ablate two training-related components that are important for enabling parallel forecasting and continuous decoding. Time noise perturbs the decoder sampling locations during training, introducing small temporal shifts that regularize the FBD and prevent overfitting to the discrete training grid (see Appendix D.1 for details). Removing time noise degrades performance from 0.725 to 0.740 MASE, supporting the interpretation that it encourages smoother continuous forecasts. We also replace causal RevIN with standard RevIN, which degrades performance to 0.738 MASE, further supporting the importance of causal normalization in our parallel forecasting setup.

**Evaluation Variants.** Finally, we evaluate two inference-time variants using the same pretrained FlowState-3M (2k) baseline, without retraining. We truncate the FBD to use only the first 128 or 64 basis functions at evaluation time. Using 128 basis functions leaves performance almost unchanged, with MASE increasing only from 0.725 to 0.726, suggesting that most forecasting-relevant information is captured by the lower-order components of the basis. Reducing the decoder further to 64 basis functions leads to a larger degradation to 0.735 MASE, indicating that higher-order basis functions still contribute to fine-grained forecast accuracy.

## 7. Conclusion

We introduce FlowState, a sampling-rate equivariant time series foundation model. To achieve equivariance, we developed a functional basis decoder (FBD), a novel component that leverages a set of basis functions to create continuous forecasts. FlowState's combination of the SSM encoder and the FBD uniquely enables the seamless adjustment of the sampling rate. To further enhance FlowState's efficiency and robustness, we propose a training scheme that through multiple parallel predictions exposes the model to diverse context lengths during training. FlowState achieves state-of-the-art performance on GIFT-Eval, outperforming all baseline models while using substantially fewer trainable parameters. Finally, we demonstrate FlowState's sampling-rate equivariance both theoretically and experimentally and our ablation studies confirm the individual and collective benefits of our proposed components.

## Limitations

Despite its state-of-the-art performance, FlowState has limitations. Firstly, to determine the appropriate scale factor for each dataset it leverages a simple heuristic, see Algorithm 1, which though worked remarkably well across a variety of datasets. Secondly, FlowState does not natively support multi-variate data or covariates. This was a deliberate simplifying design choice, allowing to be agnostic to the specific number of channels or covariates, and to process them independently in uni-variate mode. Finally, these limitations provide grounds for future work and improvements. In particular, we plan to further explore different mechanisms to automatically detect the scale factor. We will also explore information exchange mechanisms between multi-variate inputs and covariates, such as the recently introduced grouped attention over the channels (Ansari et al., 2025).

## Impact Statement

This paper presents work whose goal is to advance the field of Machine Learning. There are many potential societal consequences of our work, none of which we feel must be specifically highlighted here.

## Acknowledgements

This work has been partly funded by the EU under the Horizon Europe (HE) programme through the CloudSkin project (Grant No. 101092646).

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

# A. Additional Results

## A.1. Performance on Standard Time Series Benchmarks

We additionally evaluate FlowState on standard long-term forecasting benchmarks, including ETT (ETT1m, ETT2m, ETT1h, ETT2h), Traffic, Weather, Exchange, and Electricity. We compare against recent strong baselines: LightGTS (Wang et al., 2025a), a lightweight general forecasting model that leverages periodic structure through periodical tokenization and decoding; MOIRAI, (Woo et al., 2024) a recent time-series foundation model; and the widely used iTransformer (Liu et al., 2024b) and PatchTST (Nie et al., 2023) architectures. Table 3 shows the MSE and MAE values of the various architectures for each dataset, averaged over the forecasting lengths {96, 192, 336, 720}. FlowState performs competitively across these benchmarks and achieves the best MAE on seven out of eight datasets, indicating that its strong zero-shot performance, observed in the GIFT-Eval benchmark, transfers to standard forecasting tasks as well.

| Dataset | Metric | FlowState-10.6M (4k) | LightGTS-mini (Wang et al., 2025a) | MOIRAI-L | iTransformer (Woo et al., 2024) | PatchTST |
|---------|--------|---------------------|-----------------|----------|--------------|----------|
| ETTm1 | MSE | 0.346 | **0.327** | 0.390 | 0.407 | 0.387 |
|       | MAE | **0.354** | 0.370 | 0.389 | 0.410 | 0.400 |
| ETTm2 | MSE | 0.258 | **0.247** | 0.276 | 0.288 | 0.281 |
|       | MAE | **0.299** | 0.316 | 0.320 | 0.332 | 0.326 |
| ETTh1 | MSE | 0.393 | **0.388** | 0.510 | 0.454 | 0.469 |
|       | MAE | **0.403** | 0.419 | 0.469 | 0.448 | 0.455 |
| ETTh2 | MSE | 0.364 | **0.348** | 0.354 | 0.383 | 0.387 |
|       | MAE | 0.384 | 0.395 | **0.376** | 0.407 | 0.407 |
| Traffic | MSE | **0.381** | 0.561 | - | - | - |
|         | MAE | **0.234** | 0.381 | - | - | - |
| Weather | MSE | 0.211 | **0.208** | 0.259 | 0.258 | 0.259 |
|         | MAE | **0.236** | 0.256 | 0.275 | 0.278 | 0.281 |
| Exchange | MSE | 0.349 | **0.347** | - | - | - |
|          | MAE | **0.396** | **0.396** | - | - | - |
| Electricity | MSE | **0.155** | 0.213 | 0.188 | 0.178 | 0.216 |
|             | MAE | **0.240** | 0.308 | 0.273 | 0.270 | 0.304 |

*Table 3.* **Performance on standard benchmarks.** FlowState achieves competitive or state-of-the-art performance across a wide range of datasets. Best results are in bold and second-best are underlined.

## A.2. Qualitative Analysis of the Fixed Decoder Ablation

In Section 6.2.2 and Table 2, we ablate the functional basis decoder by replacing it with a resolution-agnostic fixed linear decoder operating on the same SSM encoder representations. This ablation preserves the sampling-rate-adjustable encoder, but removes the continuous functional decoding mechanism. This leads to a degradation from 0.725 to 0.754 MASE, indicating that the FBD contributes substantially to the overall performance of FlowState.

To better understand this degradation, we compare forecasts from the FlowState-3M baseline and the fixed-decoder ablation, both trained on 2k context length, on synthetic sine waves with different seasonalities. The results are shown in Figures 5–7. For short seasonalities, such as seasonality 8, the fixed decoder fails to adapt to the required output resolution and produces visibly distorted forecasts. In contrast, the FlowState baseline remains stable. For seasonality 24, which corresponds to the base seasonality used during training and is abundant in the pretraining corpus, both models produce reasonable forecasts. For larger seasonalities, such as seasonality 100, the fixed-decoder model captures the global trend reasonably well, but individual forecast patches exhibit local deviations. This suggests that the SSM encoder and Multi-Patch Inference can partially preserve the global forecast structure, while the absence of the FBD primarily harms local resolution adaptation within each decoded patch.

Overall, these qualitative examples support the interpretation of the ablation: the FBD is a key component for preserving

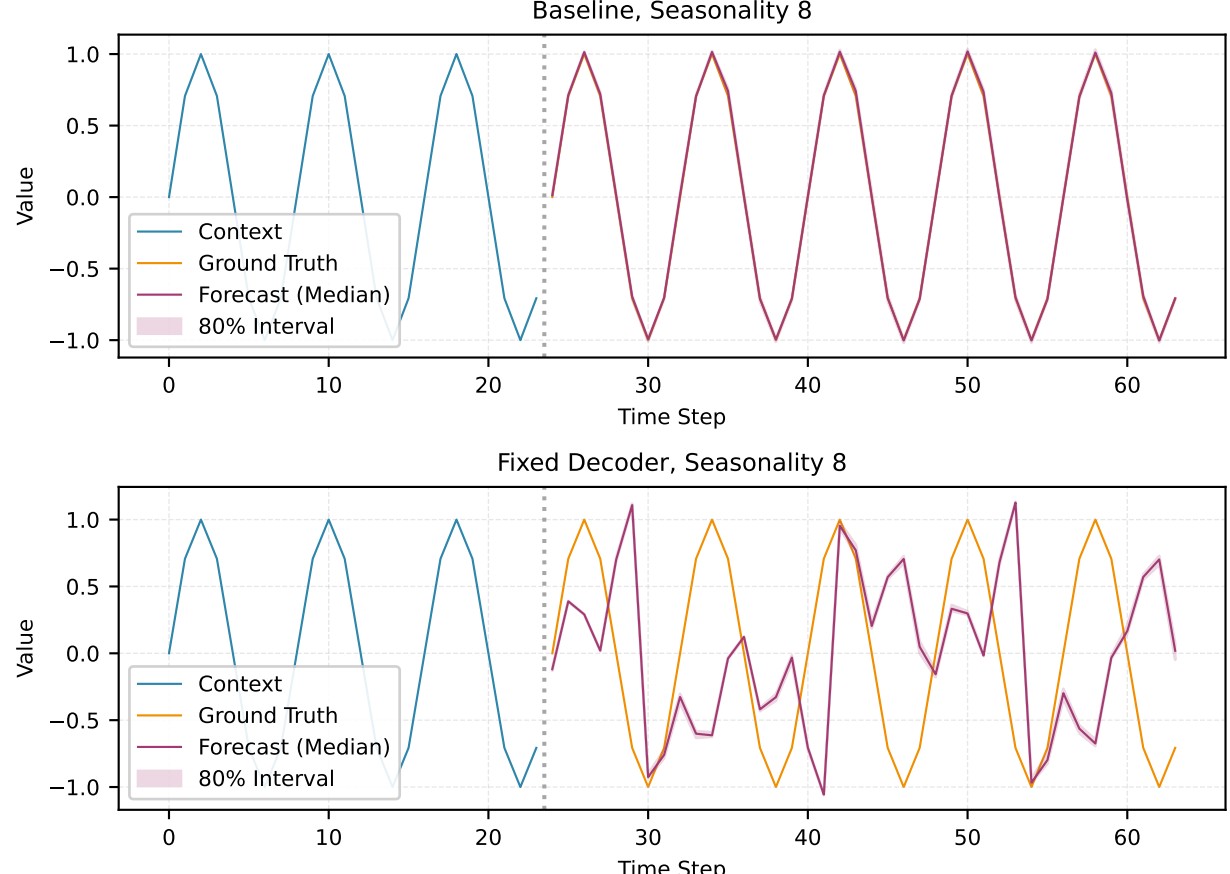

*Figure 5.* **Fixed decoder ablation for seasonality 8.** The FlowState baseline remains stable for short seasonalities, whereas the fixed linear decoder fails to adapt to the required output resolution.

sampling-rate equivariance from the encoder to the output space.

## B. Additional Details on Sampling-Rate Equivariance

### B.1. Formal Analysis of Sampling-Rate Equivariance in FlowState

We provide a formal justification of the approximate sampling-rate equivariance of FlowState in the single-layer setting. This setting captures the central mechanism: a ZOH-discretized continuous-time SSM, followed by pointwise Lipschitz operations and a rate-agnostic functional basis decoder. We then briefly discuss how the same reasoning extends to deeper encoders under additional stability and Lipschitz assumptions.

**Continuous-time SSM and ZOH discretization.** Consider the continuous-time linear state-space system

$$\dot{s}(t) = As(t) + Bu(t), \tag{17}$$
$$z(t) = Cs(t) + Du(t), \tag{18}$$

where $s(t) \in \mathbb{C}^N$ is the hidden state and $u(t) \in \mathbb{R}^H$ is the input. The SSM recurrence used in FlowState is obtained by zero-order-hold (ZOH) discretization: for a sampling interval $\Delta > 0$, the continuous input is replaced by the piecewise-constant reconstruction

$$\tilde{u}^{(\Delta)}(t) = u(t_\Delta(t)), \qquad t_\Delta(t) = \left\lfloor \frac{t}{\Delta} \right\rfloor \Delta. \tag{19}$$

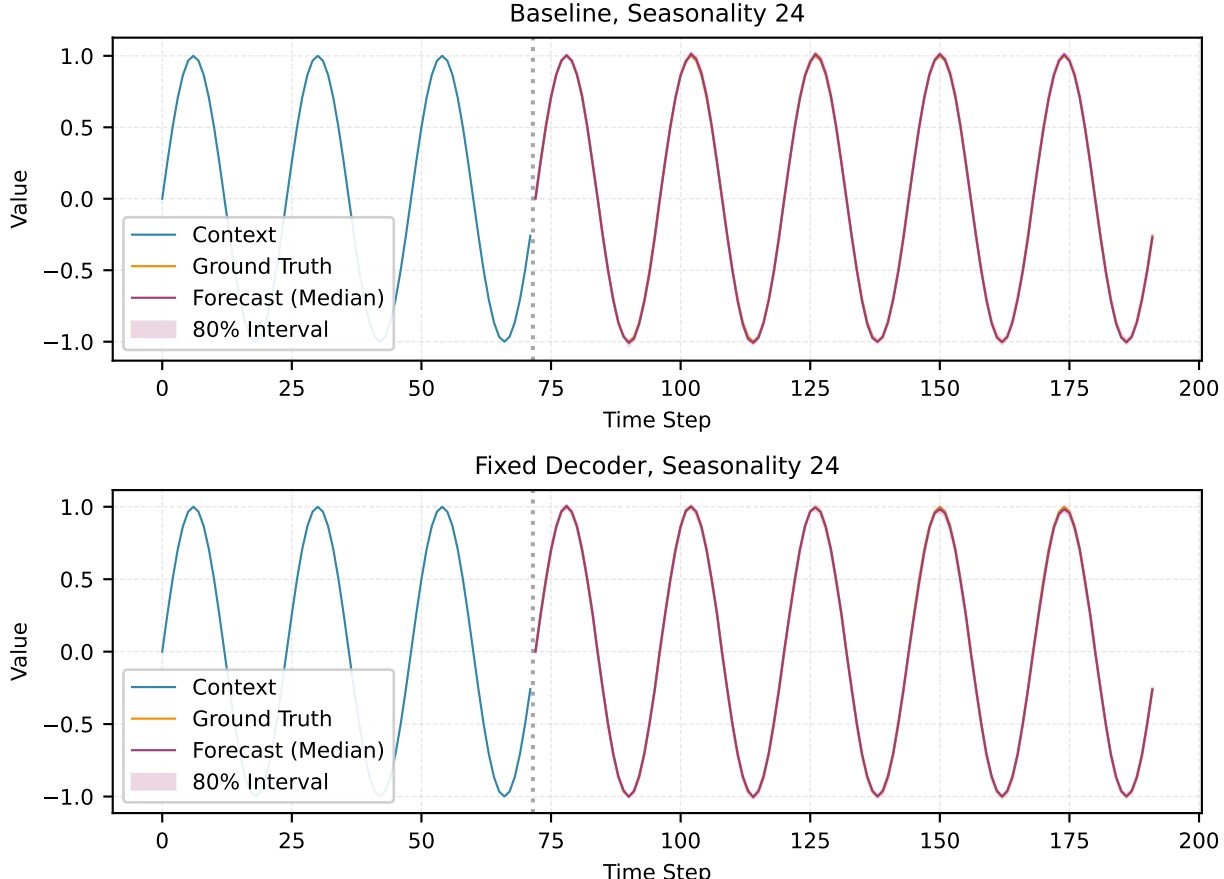

*Figure 6.* **Fixed decoder ablation for seasonality 24.** Both the FlowState baseline and the fixed decoder produce reasonable forecasts, indicating that the fixed decoder performs best near the resolution it most frequently observes during training.

The discretized SSM is by definition exactly equivalent to the sampled solution of the continuous-time system driven by $\tilde{u}^{(\Delta)}$ (Åström & Wittenmark, 2013). The only approximation relative to the original continuous signal $u(t)$ comes from this ZOH reconstruction.

**Assumptions.** We assume the following standard regularity conditions.

**Assumption B.1** (Input regularity). The input signal $u : [0, T] \rightarrow \mathbb{R}^H$ is Lipschitz continuous with constant $M$, i.e.

$$\|u(t) - u(t')\| \leq M|t - t'| \qquad \forall t, t' \in [0, T]. \tag{20}$$

**Assumption B.2** (Exponential stability). The state matrix $A$ is exponentially stable: there exists $\alpha > 0$ such that

$$\|e^{At}\| \leq e^{-\alpha t} \qquad \forall t \geq 0. \tag{21}$$

For diagonal SSMs this holds, if $\mathrm{Re}(\lambda_i) \leq -\alpha$ for all eigenvalues $\lambda_i$ of $A$.

In FlowState, this condition is ensured by parameterizing the real part of each diagonal eigenvalue as

$$\mathrm{Re}(\lambda_i) = -\exp(\theta_i), \tag{22}$$

where $\theta_i \in \mathbb{R}$ is trainable. Hence $\mathrm{Re}(\lambda_i) < 0$ for all $i$. These assumptions are mild on finite horizons and are standard for analyzing stable continuous-time SSMs under discretization.

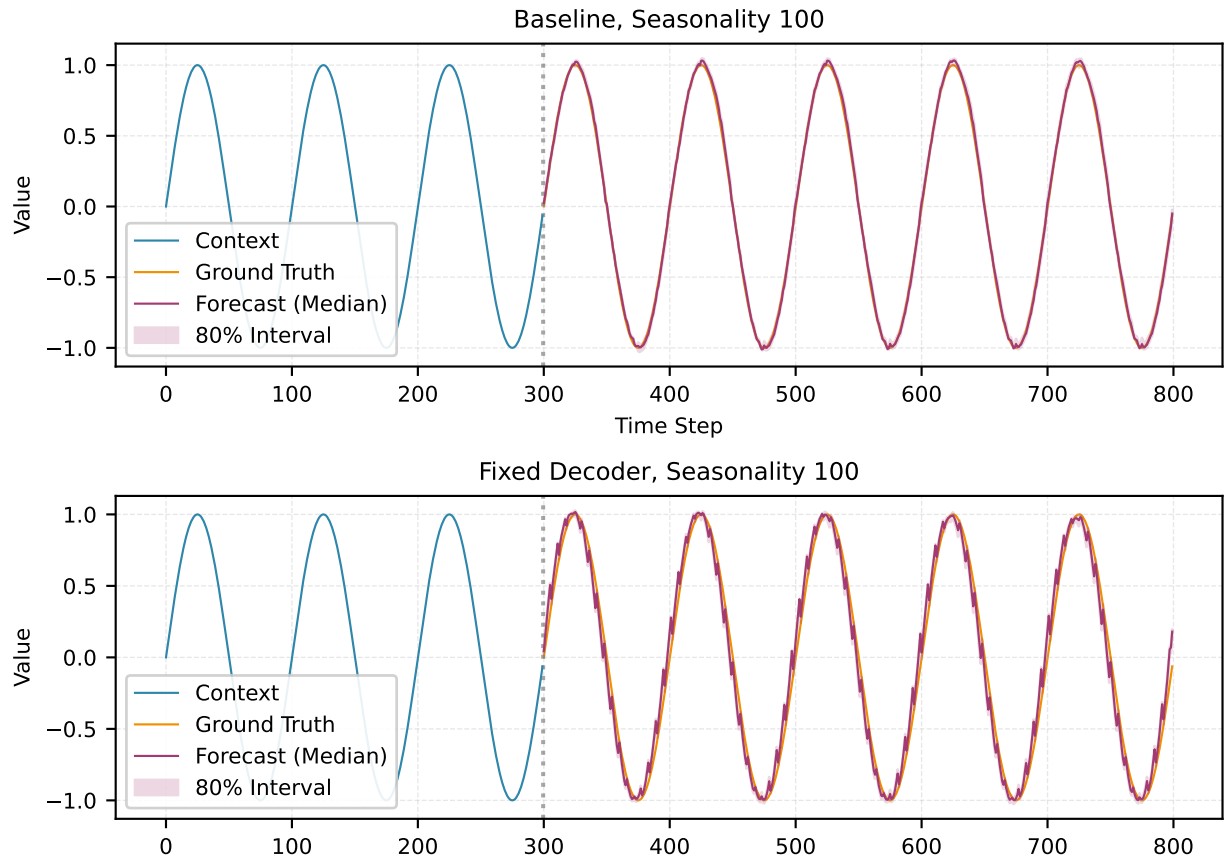

*Figure 7.* **Fixed decoder ablation for seasonality 100.** For larger seasonalities, the fixed decoder preserves the coarse global structure but introduces local deviations within individual prediction patches.

**Single-layer ZOH mismatch bound.**

**Proposition B.3** (Two-rate ZOH mismatch). *Let $s^{(\Delta)}(T)$ and $s^{(\Delta')}(T)$ denote the final hidden states obtained by driving the same continuous-time SSM with the ZOH reconstructions $\tilde{u}^{(\Delta)}$ and $\tilde{u}^{(\Delta')}$, respectively. If both systems use the same initial state, then*

$$\left\| s^{(\Delta)}(T) - s^{(\Delta')}(T) \right\| \leq \frac{\|B\|M}{\alpha} \left( 1 - e^{-\alpha T} \right) \max\{\Delta, \Delta'\}. \tag{23}$$

*Proof.* Define the mismatch

$$e(t) = s^{(\Delta)}(t) - s^{(\Delta')}(t). \tag{24}$$

Since both systems share the same dynamics and initial state,

$$\dot{e}(t) = Ae(t) + B \left( \tilde{u}^{(\Delta)}(t) - \tilde{u}^{(\Delta')}(t) \right), \qquad e(0) = 0. \tag{25}$$

By the variation-of-constants formula,

$$e(T) = \int_0^T e^{A(T-\tau)} B \left( \tilde{u}^{(\Delta)}(\tau) - \tilde{u}^{(\Delta')}(\tau) \right) d\tau. \tag{26}$$

For any $\tau \in [0, T]$,

$$\tau - \Delta < t_\Delta(\tau) \leq \tau, \qquad \tau - \Delta' < t_{\Delta'}(\tau) \leq \tau, \tag{27}$$

and hence

$$|t_\Delta(\tau) - t_{\Delta'}(\tau)| \leq \max\{\Delta, \Delta'\}. \tag{28}$$

By Lipschitz continuity of $u$,

$$\left\|\tilde{u}^{(\Delta)}(\tau) - \tilde{u}^{(\Delta')}(\tau)\right\| = \|u(t_\Delta(\tau)) - u(t_{\Delta'}(\tau))\| \leq M \max\{\Delta, \Delta'\}. \tag{29}$$

Therefore, using exponential stability,

$$\|e(T)\| \leq \int_0^T \|e^{A(T-\tau)}\| \|B\| \left\|\tilde{u}^{(\Delta)}(\tau) - \tilde{u}^{(\Delta')}(\tau)\right\| d\tau \tag{30}$$

$$\leq \|B\| M \max\{\Delta, \Delta'\} \int_0^T e^{-\alpha(T-\tau)} d\tau \tag{31}$$

$$= \frac{\|B\| M}{\alpha} \left(1 - e^{-\alpha T}\right) \max\{\Delta, \Delta'\}. \tag{32}$$

This proves the claim. $\qquad\square$

This proposition shows that the hidden state induced by the same underlying continuous signal changes only by first-order ZOH error when the sampling interval is changed.

**Pointwise operations within a FlowState layer.** The SSM block in FlowState is followed by pointwise operations, including the output gate, residual connections, normalization, and an MLP. These operations act independently at each time step and therefore do not introduce additional temporal discretization error. They may, however, amplify or attenuate the mismatch produced by the SSM state.

Let $\Phi_{\text{point}}$ denote the composition of these pointwise operations within one FlowState layer. We assume that, on the bounded set of activations encountered on the finite time horizon, $\Phi_{\text{point}}$ is Lipschitz continuous with constant $L_{\text{point}}$, i.e.

$$\|\Phi_{\text{point}}(z) - \Phi_{\text{point}}(z')\| \leq L_{\text{point}} \|z - z'\|. \tag{33}$$

This assumption is satisfied for standard neural network components with bounded parameters and bounded activations. In particular, sigmoid gates, affine maps, residual connections, MLPs with Lipschitz nonlinearities, and normalization layers with non-vanishing variance are Lipschitz on bounded domains.

Combining this Lipschitz bound with Proposition B.3, the mismatch after one complete FlowState encoder layer satisfies

$$\|\text{Enc}_\Delta(u) - \text{Enc}_{\Delta'}(u)\| \leq L_{\text{point}} \left\|s^{(\Delta)}(T) - s^{(\Delta')}(T)\right\| \tag{34}$$

$$\leq L_{\text{point}} \frac{\|B\| M}{\alpha} \left(1 - e^{-\alpha T}\right) \max\{\Delta, \Delta'\}. \tag{35}$$

Thus, for a single FlowState encoder layer, the sampling-rate mismatch remains first order in $\max\{\Delta, \Delta'\}$. The pointwise operations only change the multiplicative constant.

**Functional basis decoder.** The Functional Basis Decoder maps the final encoder representation to a continuous forecast. Writing the encoder output as coefficients $c$, the FBD constructs

$$\tilde{y}(t) = \sum_{i=1}^{n} c_i p_i(t), \qquad t \in [a, b], \tag{36}$$

where $p_i$ are fixed basis functions, such as Legendre polynomials. On any compact forecast interval $[a, b]$, these basis functions are bounded. Hence the coefficient-to- function map is linear and Lipschitz. In particular, if

$$K_p = \sup_{t \in [a,b]} \|p(t)\| < \infty, \tag{37}$$

then, for two coefficient vectors (encoder outputs) $c, c'$,

$$\|\tilde{y}_c - \tilde{y}_{c'}\|_\infty \leq K_p \|c - c'\|. \tag{38}$$

Sampling the continuous forecast at interval $\Delta_F$ simply evaluates $\tilde{y}(t)$ on the corresponding output grid. Therefore, the decoder can change the output sampling rate without changing the underlying continuous forecast.

**End-to-end single-layer sampling-rate equivariance.** We now combine the SSM bound, the pointwise encoder operations, and the Functional Basis Decoder. Let

$$\mathcal{F}_\Delta(u) = \text{FBD}_\Delta\left(\text{Enc}_\Delta(u)\right) \tag{39}$$

denote FlowState applied to an input sampled with interval $\Delta$, with the resulting continuous forecast evaluated on the same $\Delta$-grid. More generally, $\text{FBD}_{\Delta'}$ denotes evaluation of the continuous FBD forecast on a grid with spacing $\Delta'$.

We compare predictions on the same output grid $\Delta'$:

$$\mathcal{F}_{\Delta'}(u) = \text{FBD}_{\Delta'}\left(\text{Enc}_{\Delta'}(u)\right) \tag{40}$$

and

$$\text{FBD}_{\Delta'}\left(\text{Enc}_\Delta(u)\right). \tag{41}$$

The first processes the input at sampling interval $\Delta'$, while the second processes the same underlying signal at sampling interval $\Delta$, but evaluates the forecast on the $\Delta'$-grid.

Using the Lipschitz continuity of the FBD and Eq. (35), we obtain

$$\|\mathcal{F}_{\Delta'}(u) - \text{FBD}_{\Delta'}\left(\text{Enc}_\Delta(u)\right)\| \tag{42}$$
$$= \|\text{FBD}_{\Delta'}\left(\text{Enc}_{\Delta'}(u)\right) - \text{FBD}_{\Delta'}\left(\text{Enc}_\Delta(u)\right)\| \tag{43}$$
$$\leq C_{\text{FBD}} \|\text{Enc}_{\Delta'}(u) - \text{Enc}_\Delta(u)\| \tag{44}$$
$$\leq C_{\text{FBD}} L_{\text{point}} \frac{\|B\| M}{\alpha} \left(1 - e^{-\alpha T}\right) \max\{\Delta, \Delta'\}. \tag{45}$$

Therefore, for a single-layer FlowState encoder, the equivariance error satisfies

$$\|\mathcal{F}_{\Delta'}(u) - \text{FBD}_{\Delta'}\left(\text{Enc}_\Delta(u)\right)\| \leq C \max\{\Delta, \Delta'\}, \tag{46}$$

with

$$C = C_{\text{FBD}} L_{\text{point}} \frac{\|B\| M}{\alpha} \left(1 - e^{-\alpha T}\right). \tag{47}$$

Thus, FlowState is approximately sampling-rate equivariant in the single-layer setting: changing the input sampling rate changes the prediction on a fixed output grid only through the encoder's ZOH discretization error, and this error vanishes as

$$\max\{\Delta, \Delta'\} \to 0. \tag{48}$$

**Interpretation of the bound.** The bound in Eq. (45) separates the source of the sampling-rate equivariance error from its possible amplification by the model. The central term is $M \max\{\Delta, \Delta'\}$, which upper bounds the worst-case mismatch between two ZOH reconstructions of the same continuous input. Intuitively, if the input changes only slowly within one sampling interval, then this mismatch is small, and the induced equivariance error is small as well.

The remaining factors in the bound are model-dependent constants. They describe how the SSM dynamics, the pointwise neural components, and the FBD may amplify the initial mismatch in a worst-case analysis. In practice, this worst-case amplification need not be realized. In particular, pretraining FlowState across multiple sampling rates and using time-noise during training encourages the model to produce smooth, rate-robust representations. Thus, the non-SSM components may also attenuate, rather than amplify, the discretization-induced mismatch.

**Remark on multiple encoder layers.** The formal bound above is stated for a single SSM layer for clarity. In the full FlowState encoder, later SSM layers receive inputs that are themselves produced by earlier layers and may therefore already differ across sampling rates. A full multi-layer proof requires tracking both the propagated mismatch from previous layers and the fresh ZOH discretization error introduced at each subsequent SSM layer.

For a later layer $\ell > 1$, let $u_{\ell,\Delta}(t)$ and $u_{\ell,\Delta'}(t)$ denote the continuous-time input trajectories to layer $\ell$ induced by processing the signal at sampling intervals $\Delta$ and $\Delta'$, respectively. Define the propagated mismatch from the previous layer as

$$E_{\ell-1} = \sup_t \|u_{\ell,\Delta}(t) - u_{\ell,\Delta'}(t)\|. \tag{49}$$

Assume additionally that $u_{\ell,\Delta}$ is Lipschitz with constant $M_\ell$ on the finite horizon. Then, for any $\tau$,

$$\|u_{\ell,\Delta}(t_\Delta(\tau)) - u_{\ell,\Delta'}(t_{\Delta'}(\tau))\| \tag{50}$$
$$\leq \|u_{\ell,\Delta}(t_\Delta(\tau)) - u_{\ell,\Delta}(t_{\Delta'}(\tau))\| + \|u_{\ell,\Delta}(t_{\Delta'}(\tau)) - u_{\ell,\Delta'}(t_{\Delta'}(\tau))\| \tag{51}$$
$$\leq M_\ell \max\{\Delta, \Delta'\} + E_{\ell-1}. \tag{52}$$

Thus, compared to the first-layer proof, the input mismatch to layer $\ell$ contains two terms: the fresh ZOH mismatch $M_\ell \max\{\Delta, \Delta'\}$, and the propagated mismatch $E_{\ell-1}$ from previous layers.

Under uniform stability of all SSM layers and uniform Lipschitz bounds on the pointwise components, this yields a recursion of the form

$$E_\ell \leq L_\ell E_{\ell-1} + C_\ell \max\{\Delta, \Delta'\}, \qquad E_0 = 0, \tag{53}$$

where $E_\ell$ denotes the sampling-rate mismatch after layer $\ell$, and $L_\ell, C_\ell$ are layer-dependent constants independent of $\Delta$ and $\Delta'$. Unrolling this recursion gives

$$E_K = O(\max\{\Delta, \Delta'\}), \tag{54}$$

with a larger model-dependent constant. Therefore, additional layers affect the prefactor of the equivariance error, but not its first-order dependence on the sampling interval.

## B.2. Empirical Validation of Sampling-Rate Equivariance

In the previous section we proved the sample-rate equivariance of FlowState. A central component of this equivariance is the approximate invariance across time steps of the SSM encoder. We investigated this invariance empirically by providing contexts from ETT1m and ETT1h to FlowState and measure the correlation (negative cosine similarity) between the hidden states of the last encoding layer. In particular, our experiment can be outlined as

$$\boldsymbol{h}_{L,\text{ETT1m}}^N = \text{SSM}_{\Delta_E^l=0.25}(\boldsymbol{x}_{\text{ETT1m}})$$
$$\boldsymbol{h}_{L,\text{ETT1h}}^N = \text{SSM}_{\Delta_E^l=1}(\boldsymbol{x}_{\text{ETT1h}}),$$

where $\Delta_E^l = 0.25$ and $\Delta_E^l = 1$ are the typical scale factors for the ETT1m and ETT1h datasets. According to our explanation in Section 4, these two encodings should be approximately similar to each other, i.e., $\boldsymbol{h}_{L,\text{ETT1m}}^N \approx \boldsymbol{h}_{L,\text{ETT1h}}^N$. This is only valid if the input context $\boldsymbol{x}_{\text{ETT1m}}$ and $\boldsymbol{x}_{\text{ETT1h}}$ are from the same underlying process, but just sampled with different rates.

Figure 8a shows the correlations of $\boldsymbol{h}_{L,\text{ETT1m}}^N$ and $\boldsymbol{h}_{L,\text{ETT1h}}^N$ for different contexts from ETT1m and ETT1h. The Figure is divided into four quadrants. The top left quadrant shows the correlations of the encodings of the ETT1m data. One can see that the encodings are strongly correlated to themselves, i.e., have high values along the diagonal. On the other hand, the data naturally shows some cross-correlation, because the individual contexts from ETT1m are partly overlapping. The bottom right quadrant shows the correlations of the ETT1h dataset. The correlation pattern is similar to the one of the ETT1m dataset. The top right and bottom left quadrant show the correlation between the encodings of ETT1m and ETT1h, and ETT1h and ETT1m, respectively. One can see that the correlation pattern looks very similar to the pattern of the individual datasets themselves. In particular, the values on the diagonal, showing the correlation of the encoding for the ETT1m sample with the corresponding subsampled ETT1h context. In contrast, in Figure 8b we modify the ETT1h data such that there is no temporal correlation between ETT1m and ETT1h anymore. As one can see, while the correlation pattern within the individual datasets remain, there are no significant correlations across the datasets.

## C. Pretraining data

Table 4 and 5 list the individual datasets utilized from the GIFT-Eval-Pretrain corpus, and Table 6 lists the individual datasets utilized from the Chronos Pretraining corpus.

| Dataset | Domain | Frequency | #Time Series | # Obs. |
|---|---|---|---|---|
| BDG-2 Panther | Energy | H | 105 | 919,800 |
| BDG-2 Fox | Energy | H | 135 | 2,324,568 |
| BDG-2 Rat | Energy | H | 280 | 4,728,288 |
| BDG-2 Bear | Energy | H | 91 | 1,482,312 |
| Low Carbon London | Energy | H | 713 | 9,543,348 |
| SMART | Energy | H | 5 | 95,709 |
| IDEAL | Energy | H | 219 | 1,265,672 |
| Sceaux | Energy | H | 1 | 34,223 |
| Borealis | Energy | H | 15 | 83,269 |
| Buildings900K | Energy | H | 1,792,328 | 15,702,590,000 |
| CMIP6 | Climate | 6H | 1,351,680 | 1,973,453,000 |
| ERA5 | Climate | H | 245,760 | 2,146,959,000 |
| Azure VM Traces 2017 | CloudOps | 5T | 159,472 | 885,522,908 |
| Borg Cluster Data 2011 | CloudOps | 5T | 143,386 | 537,552,854 |
| Wiki-Rolling | Web | D | 47,675 | 40,619,100 |
| M5 | Sales | D | 30,490 | 58,327,370 |
| PEMS03 | Transport | 5T | 358 | 9,382,464 |
| PEMS04 | Transport | 5T | 307 | 5,216,544 |
| PEMS07 | Transport | 5T | 883 | 24,921,792 |
| PEMS08 | Transport | 5T | 170 | 3,035,520 |
| PEMS Bay | Transport | 5T | 325 | 16,937,700 |
| Los-Loop | Transport | 5T | 207 | 7,094,304 |
| Beijing Subway | Transport | 30T | 276 | 248,400 |
| SHMetro | Transport | 15T | 288 | 1,934,208 |
| HZMetro | Transport | 15T | 80 | 146,000 |
| Q-Traffic | Transport | 15T | 45,148 | 264,386,688 |
| Subseasonal | Climate | D | 862 | 14,097,148 |
| Subseasonal Precipitation | Climate | D | 862 | 9,760,426 |
| GEF12 | Energy | H | 20 | 788,280 |
| GEF14 | Energy | H | 1 | 17,520 |
| GEF17 | Energy | H | 8 | 140,352 |
| PDB | Energy | H | 1 | 17,520 |
| Spanish | Energy | H | 1 | 35,064 |
| BDG-2 Hog | Energy | H | 24 | 421,056 |
| BDG-2 Bull | Energy | H | 41 | 719,304 |
| BDG-2 Cockatoo | Energy | H | 5 | 17,544 |
| ELF | Energy | H | 1 | 21,792 |

*Table 4.* Datasets contained in GIFT-Eval-Pretrain for FlowState. Table adopted from Aksu et al. (2024)

| Dataset | Domain | Frequency | #Time Series | # Obs. |
|---|---|---|---|---|
| Solar Power | Energy | 4S | 1 | 7,397,222 |
| Oikolab Weather | Climate | H | 8 | 800,456 |
| Elecdemand | Energy | 30T | 1 | 17,520 |
| Covid Mobility | Transport | D | 362 | 148,602 |
| Kaggle Web Traffic Weekly | Web | W | 145,063 | 16,537,182 |
| Extended Web Traffic | Web | D | 145,063 | 370,926,091 |
| M1 Yearly | Econ/Fin | Y | 106 | 3,136 |
| M1 Quarterly | Econ/Fin | Q | 198 | 9,854 |
| M1 Monthly | Econ/Fin | M | 617 | 44,892 |
| M3 Yearly | Econ/Fin | Y | 645 | 18,319 |
| M3 Quarterly | Econ/Fin | Q | 756 | 37,004 |
| M3 Monthly | Econ/Fin | M | 1,428 | 141,858 |
| M3 Other | Econ/Fin | Q | 174 | 11,933 |
| NN5 Daily | Econ/Fin | D | 111 | 81,585 |
| NN5 Weekly | Econ/Fin | W | 111 | 11,655 |
| Tourism Yearly | Econ/Fin | Y | 419 | 11,198 |
| Tourism Quarterly | Econ/Fin | Q | 427 | 39,128 |
| Tourism Monthly | Econ/Fin | M | 366 | 100,496 |
| Traffic Weekly | Transport | W | 862 | 82,752 |
| Traffic Hourly | Transport | H | 862 | 14,978,112 |
| Australian Electricity Demand | Energy | 30T | 5 | 1,153,584 |
| Sunspot | Nature | D | 1 | 73,894 |
| Vehicle Trips | Transport | D | 329 | 32,512 |
| Weather | Climate | D | 3,010 | 42,941,700 |
| FRED MD | Econ/Fin | M | 107 | 76,612 |
| Pedestrian Counts | Transport | H | 66 | 3,130,762 |
| Bitcoin | Econ/Fin | D | 18 | 74,824 |
| KDD Cup 2022 | Energy | 10T | 134 | 4,727,519 |
| GoDaddy | Econ/Fin | M | 3,135 | 128,535 |
| Favorita Sales | Sales | D | 111,840 | 139,179,538 |
| Favorita Transactions | Sales | D | 54 | 84,408 |
| Residential Load Power | Energy | T | 271 | 145,994,559 |
| Residential PV Power | Energy | T | 233 | 125,338,950 |
| CDC Fluview ILINet | Healthcare | W | 75 | 63,903 |
| CDC Fluview WHO NREVSS | Healthcare | W | 74 | 41,760 |
| Project Tycho | Healthcare | W | 1,258 | 1,377,707 |
| Wind Power | Energy | 4S | 1 | 7397147 |

*Table 5.* Datasets contained in GIFT-Eval-Pretrain continued. Table adopted from Aksu et al. (2024).

## C.1. Data balancing

As our pretraining data corpus consists of many individual datasets, a question arises with which probability each dataset in the corpus is sampled. In particular, we observed that only a few datasets are dominant and much larger than the others, which may adversely affect training. Some TSFMs also distinguish between the dataset domain or the sampling frequency to balance the training data. However, we did not follow that approach, but instead we sample the individual time series from the datasets depending on their size. For example, if a dataset is very large, i.e., contains a lot of data points, we subsample this dataset and present fewer time series of that dataset to the model than for a smaller dataset.

## C.2. Preprocessing

We perform one minor preprocessing step on the individual time series of the datasets, before we provide them to FlowState. If a time series contains too much missing data, typically denoted with NaN values, we remove them by splitting the time series into smaller pieces that don't contain as many NaN values anymore. Although this avoids dealing with excessive NaN values, it may result in creating more short time series.

## C.3. Synthetic Data and Data Augmentations

We augment each data corpus with synthetic data that is generated in a manner inspired by the CauKer method presented in (Xie et al., 2025). We adopted the same filter banks and approach to generate the time series, but we re-implemented the method using GPyTorch, for GPU acceleration. In addition to the time series, we record the respective sampling rates that are contained in the time series, which we then later use to train FlowState.

On top of data from the real corpus, as well as on the synthetic data, we applied augmentations inspired by Auer et al. (2025). In particular, we used a modified amplitude modulation, the censor augmentation, but did not use the augmentation using the irregular spikes.

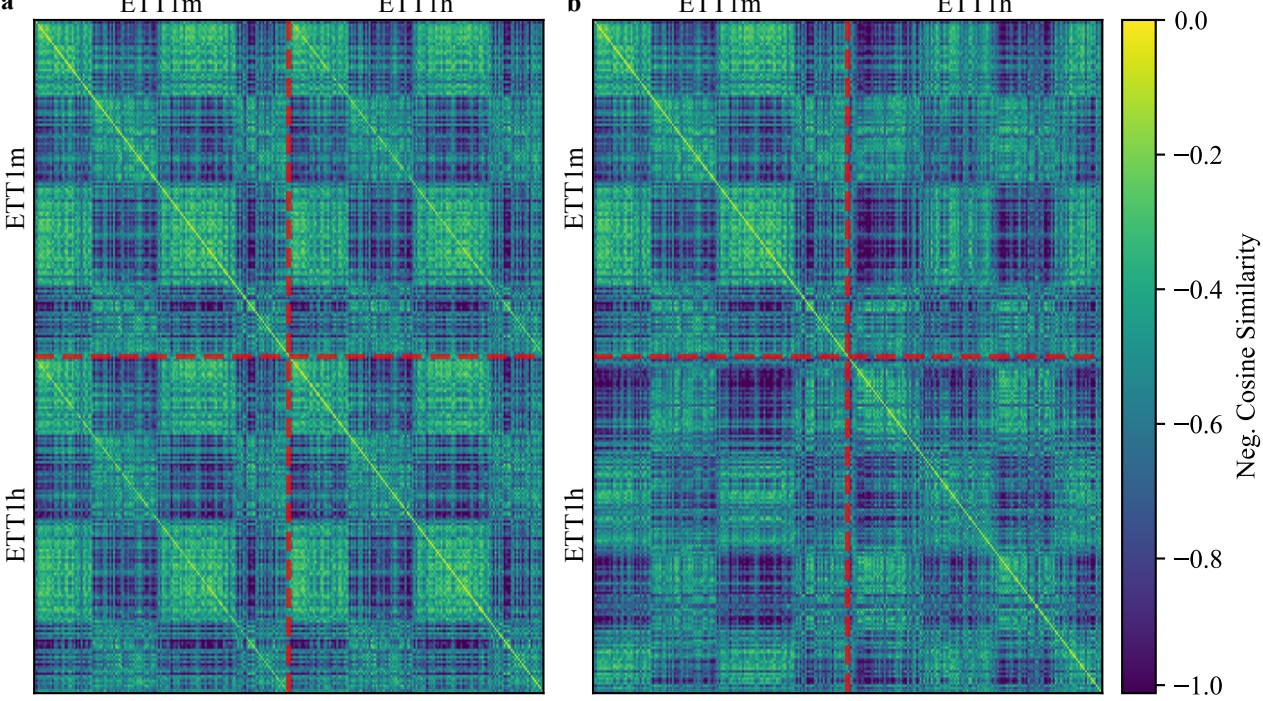

*Figure 8.* **Correlation of SSM encodings from FlowState for ETT1m and ETT1h**. This Figure illustrates the correlation between the encodings of the final SSM layer of the SSM encoder of FlowState. **a** The input contexts from the ETT1m and ETT1h data are temporally aligned, i.e., the data from ETT1h is a subsampled version of the data from ETT1m. **b** The input contexts from the ETT1m and ETT1h data are temporally mis-aligned, i.e., the data from ETT1h is a subsampled version of the data from ETT1m, but it is flipped along the time axis, such that there is no temporal alignment.

| Benchmark | Dataset | Domain | Frequency | #Time Series | Avg. length |
|---|---|---|---|---|---|
| Chronos | Mexico City Bikes | Transport | H | 494 | 78313 |
| Chronos | Brazilian Cities Temperature | Nature | M | 12 | 757 |
| Chronos | Spanish Energy and Weather | Energy | H | 66 | 35064 |
| Chronos | USHCN | Nature | D | 6090 | 38653 |
| Chronos | Weatherbench (Hourly) | Nature | H | 225280 | 350639 |
| Chronos | Weatherbench (Daily) | Nature | D | 225280 | 14609 |
| Chronos | Weatherbench (Weekly) | Nature | W | 225280 | 2087 |
| Chronos | Wiki Daily (100k) | Web | D | 100000 | 2741 |
| Chronos | Wind Farms (Hourly) | Energy | H | 100000 | 8514 |
| Chronos | Wind Farms (Daily) | Energy | D | 100000 | 354 |
| Chronos | London Smart Meters | Energy | 30T | 5560 | 29951 |
| Chronos | Pedestrian Counts | Transport | H | 66 | 47459 |
| Chronos | Rideshare | Transport | H | 2340 | 541 |
| Chronos | Uber TLC (Daily) | Transport | D | 262 | 181 |

*Table 6.* Chronos data contained in the TiRex pretraining data. Table adopted from Auer et al. (2025).

## D. Additional training details

### D.1. Pretraining

We pretrained our model using the quantile loss function with quantile levels ranging from 0.1 to 0.9 in increments of 0.1, resulting in $Q = 9$ levels. The resulting loss is:

$$L_Q = \frac{1}{T \cdot Q} \sum_{t=0}^{T-1} \sum_{q=0.1}^{0.9} \begin{cases} (1-q) \cdot |\hat{\boldsymbol{y}}_t^q - \boldsymbol{y}_t|, & \text{if } \hat{\boldsymbol{y}}_t^q < \boldsymbol{y}_t \\ q \cdot |\hat{\boldsymbol{y}}_t^q - \boldsymbol{y}_t|, & \text{else} \end{cases} \tag{55}$$

While datasets are treated as univariate series, and therefore predictions always have only one channel, each forecasting timestep consists of $Q$ quantiles, where $\hat{\boldsymbol{y}}_t^q$ is the $q$'th quantile forecast at timestep $t$.

| Hyperparameter | Value |
|---|---|
| Context Length L | 4096 |
| Min. Context Length $L_{min}$ | 20 |
| Learning Rate | 1.5e-4 |
| SSM Learning Rate | 5e-5 |
| Optimizer | AdamW |
| Batch Size | 64 |
| Warmup Steps | ≈38k |
| Total Training Steps | ≈ 800k |
| Weight Decay | 0.05 |
| Gradient Clipping | 5.0 |
| Scheduler | Cosine Annealing |
| Dropout Rate | 0.0 |
| Seed | 0 |
| Time Noise ($\tau$) | 0.1 |
| Base Seasonality (B) | 24 |
| Loss Function | Quantile Loss |

*Table 7.* Training hyperparameters used during pretraining. Following Smith et al. (2023), a smaller learning rate and no weight decay was used for SSM parameters $A$, $B$, and $\Delta$.

We use no dropout, but a novel regularization technique inside FBD, called time noise. The purpose of time noise is to ensure that we learn smooth functions, and do not overfit on the specific time points present in our training data corpus. For

---
**Algorithm 1 GetSeason**: Determining the Seasonality

---
  **Input:** $SamplingInterval$, $domain$
  **if** $domain$ in ["Transport", "Healthcare", "Web/CloudOps", "Sales"] **then**
    $HasWeekly = True$
  **end if**
  **if** $SamplingInterval < 1$ min **then**
    **Return:** 1 hour/$SamplingInterval$ {Sub-hourly group}
  **else if** $SamplingInterval < 1$ day **then**
    **Return:** 1 day/$SamplingInterval$ {Sub-daily group}
  **else if** $SamplingInterval < 1$ week **and** $HasWeekly$ **then**
    **Return:** 1 week/$SamplingInterval$ {Sub-weekly group}
  **else if** $SamplingInterval < 1$ year **then**
    **Return:** 1 year/$SamplingInterval$ {Sub-yearly group}
  **else**
    **Return:** 4 {Yearly}
  **end if**

---

this purpose, we add Gaussian noise on top of our time vector used for quantization. This Gaussian noise is always relative to the scale factor $s_\Delta$ and therefore to the step size of our time vector used for discretizing continuous functions. Given a continuous forecast $\tilde{y}$, linearly sampled at a discrete time vector $\boldsymbol{t}_\Delta$ with sampling interval $\Delta$, time noise adapts $\boldsymbol{t}_\Delta$ in the following way:

$$\tilde{\boldsymbol{t}}_\Delta = \boldsymbol{t}_\Delta + \boldsymbol{\epsilon}, \tag{56}$$

where $\boldsymbol{\epsilon}$ is a vector consisting of independently drawn zero-mean Gaussian noise with standard deviation of $\tau \cdot \Delta$, and $\tau$ is the time noise hyperparameter set to $0.1$ for all experiments.

For all experiments PyTorch random seed $0$ was used, and experiments were executed once, because we did not observe a major performance variation. Further implementation details are summarized in Table 7.

### D.2. Evaluation

All of our GIFT-Eval Leaderboard data is in accordance with the official leaderboards as of Jan 28th, 2026.

#### D.2.1. ADDITIONAL INFORMATION ABOUT SCALE FACTOR SELECTION

As explained in Section 6.1.2 of the main text, $s_\Delta$ is determined by the seasonality of each dataset, and by the base seasonality ($B = 24$ for all experiments). If seasonality is known to be $N$, $s_\Delta = B/N$.

For GIFT-Eval, the seasonality was determined by a simple mapping from sampling rate and domain to seasonality. The domain was only necessary to determine whether the dataset contains a weekly cycle or not. This is the case for data depending on the human work cycle, but not for other datasets, such as for example from domain "Nature". We determined that domains: "Transport", "Healthcare", "Web/CloudOps" and "Sales" have a weekly cycle, and others don't. With this in mind, the seasonality was determined by the number of time steps within the next larger season according to the Algorithm 1.

The only exception was "bizitobs_l2c", which did not appear to have a daily cycle, and therefore we directly went for the next larger cycle, which was the weekly cycle.

#### D.2.2. AUTOMATIC SCALE-FACTOR SELECTION

In the main experiments, the scale factor $s_\Delta$ is selected from prior knowledge of the dataset seasonality. To reduce the dependency on externally provided seasonality information, we additionally evaluate FlowState-3M (2k) with an automatic scale-factor selection procedure.

For each provided context $\mathbf{X} \in \mathbb{R}^{L \times c}$, we estimate the dominant seasonality by sweeping over candidate seasonalities $s \in \{s_{\min}, \ldots, s_{\max}\}$. For each candidate $s$, we compute the seasonality error

$$\mathbb{E}_{\text{season}}(\mathbf{X}, s) = \text{mean}\left(|\mathbf{X}_{1:L-s} - \mathbf{X}_{s+1:L}|\right),$$

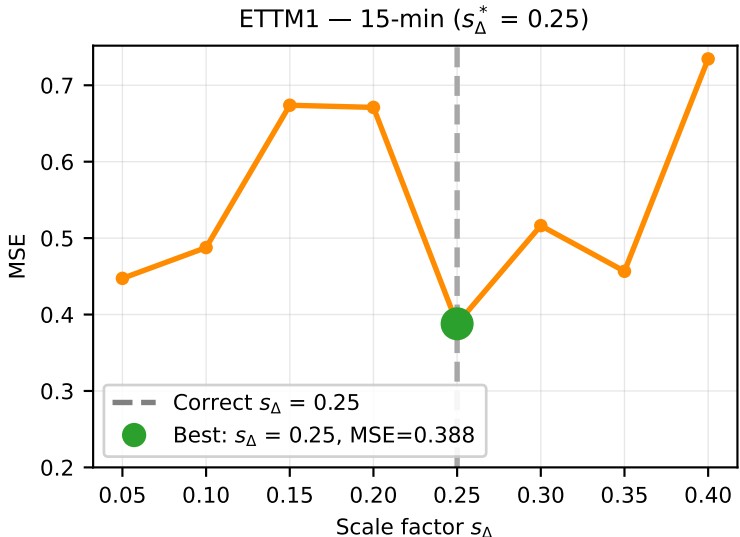

*Figure 9.* Scale-factor sensitivity analysis on ETTm1.

where $L$ is the context length. Low values of $\mathbb{E}_{\text{season}}(\mathbf{X}, s)$ indicate that the sequence is similar to itself after a shift of $s$ steps, and therefore suggest a plausible seasonality.

Given the resulting error curve over candidate seasonalities, we select the seasonality as the first sufficiently prominent local minimum. In practice, we apply SciPy's peak-finder[3] and select the first detected local minima above a fixed prominence threshold. The selected seasonality is then converted into a scale factor using the same rule as in Section 6.1.2.

Compared to the scale-factor selection based on dataset-level seasonality information, this automatic procedure slightly degrades GIFT-Eval performance for FlowState-3M (2k), from MASE 0.725 and CRPS 0.502 to MASE 0.746 and CRPS 0.521. This indicates that automatic scale selection is feasible, but still trails the setting where sampling-rate or seasonality information is available.

Inspecting the individual GIFT-Eval sub-tasks shows that the degradation is mainly dominated by a small number of datasets with short context lengths, where reliable seasonality estimation is difficult. For most datasets, automatic scale-factor selection leads to only a moderate decrease, or no decrease, in forecasting performance.

### D.2.3. SCALE FACTOR SENSITIVITY

To quantify the sensitivity of FlowState to the scale factor, we evaluate the effect of varying the scale factor $s_\Delta$ on the commonly used ETTm1 dataset. As shown in Figure 9, the expected scale factor $s_\Delta = 0.25$ for ETTm1 with seasonality of 96 achieves the best performance. Performance degrades for misaligned scale factors, indicating that appropriate temporal scaling is important for sampling-rate equivariant forecasting.

### D.3. Hyperparameters

Model hyperparameters can be found in Table 8. No extensive hyperparameter search has been performed. The only architectural difference between FlowState-10.6M and FlowState-18.6M is that the 18.6M version uses a larger two-layer gated MLP with expansion factor 2, whereas the 10.6M version uses the single-layer self-gated MLP described in Equation 3.

### D.4. Training infrastructure

As shown in the tables of the main manuscript, FlowState is the smallest model compared to contemporary TSFMs. Therefore, it also has a small memory footprint and can easily fit onto a single GPU. However, in order to expedite the training process, we utilized a GPU-based machine with 4x NVIDIA A100 80GB, a 2.4GHz Intel Xenon CascadeLake CPU

---

[3]https://docs.scipy.org/doc/scipy/reference/generated/scipy.signal.find_peaks.html

| Component | FlowState-3M | FlowState-10.6M/ 18.6M |
|-----------|:------------:|:----------------------:|
| # Layers | 6 | 6 |
| Hidden Size | 256 | 512 |
| State Dimension | 256 | 512 |
| # Basis Functions | 256 | 256 |

*Table 8.* Model hyperparameters for FlowState variants.

and 1.2TB of main memory. We trained FlowState-10.6M using all 4 GPUs for about 48 hours and FlowState-3M on 2 GPUs for 40 hours.

