# OpenReview forum: "FlowState: Sampling-Rate‑Equivariant Time‑Series Forecasting"
_ICML.cc/2026/Conference — ICML 2026 regular_

### Official Review · Reviewer_wHTS · 2026-02-15

**Soundness:** 3
**Presentation:** 3
**Significance:** 3
**Originality:** 2
**Overall Recommendation:** 4
**Confidence:** 3

**Summary:**

FlowState is a novel time-series foundation model based on state space models (SSMs) that achieves sampling-rate equivariance. The architecture features an SSM encoder that processes input sequences without patching or quantization, producing sampling-rate-invariant hidden representations, and a functional basis decoder (FBD) that interprets these representations as coefficients of continuous basis functions, enabling forecasts at arbitrary resolutions. A parallel training scheme concurrently optimizes predictions from multiple context lengths, enhancing robustness and generalization. FlowState demonstrates strong zero-shot performance on the GIFT-Eval benchmark, ranking second among public models while being significantly more parameter-efficient—the 10.6M-parameter variant achieves competitive results against models up to 20× larger. The model’s equivariance property allows seamless adaptation to varying temporal resolutions, and its continuous formulation supports flexible forecast horizons. This work highlights the potential of SSM-based architectures for building lightweight yet powerful foundation models for time-series forecasting.

**Compliance With Llm Reviewing Policy:**

Affirmed.

**Key Questions For Authors:**

The determination of scaling factors s_Delta relies on a heuristic based on pre-defined seasonality (Appendix Algorithm 2). Could the model be adapted to learn these factors automatically from data during inference, rather than requiring external seasonality estimates? A positive answer would significantly strengthen the model's claim as a true foundation model.

The theoretical equivariance error bound relies on the ZOH discretization error diminishing as sampling intervals approach zero. For coarsely sampled data (e.g., daily or weekly), this error is non-negligible. Could you provide a quantitative ablation study measuring the equivariance error (e.g., MSE) across multiple sampling rates to empirically validate the robustness claims?

The parallel training scheme depends critically on strictly causal normalization to prevent information leakage. Could you provide implementation details of the causal normalization mechanism and include ablation studies demonstrating its effectiveness? This would address concerns about potential training artifacts.

The decoder draws inspiration from HiPPO's reconstruction approach but adapts it for forecasting. Could you clarify whether the basis functions are fixed (e.g., Legendre polynomials) or learned during training? This would help delineate the novelty relative to prior work.

**Limitations:**

No. The authors provide a brief ethical statement but do not adequately discuss technical limitations. They should explicitly acknowledge the dependence on heuristic seasonality estimation, the approximate nature of equivariance for coarsely sampled data, and the lack of automatic adaptation to unseen frequencies. Adding a dedicated limitations section would strengthen transparency.

**Strengths And Weaknesses:**

Strengths:

Novel and Well-Motivated Architecture: The combination of an SSM encoder with a Functional Basis Decoder offers a principled solution to sampling-rate equivariance, a key practical challenge in real-world time series forecasting.

Strong Empirical Performance: FlowState achieves state-of-the-art results on the GIFT-Eval benchmark, ranking second among public models. Its parameter efficiency (10.6M vs. TimesFM's 200M) is particularly noteworthy.

Elegant Theoretical Framing: The paper provides a clear theoretical justification for the model's approximate sampling-rate equivariance, grounding the design in established concepts like zero-order hold discretization.

Weaknesses:

Dependence on Seasonality Heuristics: Although the model's equivariance is theoretically sound, its practical deployment relies on a significant heuristic. The scaling factors are not learned automatically from data but are set based on a pre-defined base seasonality and an external estimate of each dataset's dominant seasonality. This reintroduces the dataset-specific tuning that the model aims to avoid. For a true foundation model, handling varying frequencies should be fully automatic; the current approach assumes the user knows the seasonality in advance, which may not be available or well-defined for non-periodic signals.

Limited Analysis of Equivariance Error: The paper proves that the equivariance error is bounded by the discretization error and diminishes as the sampling interval approaches zero. However, for real-world, coarsely sampled data such as hourly or daily observations, this error is non-negligible. The empirical demonstration is qualitative and shows only a single example. A more rigorous, quantitative ablation study measuring the equivariance error across a range of sampling rates and datasets is missing. Such analysis would better characterize the model's robustness when the underlying assumptions are violated.

Parallel Forecast Training and Potential Leakage: The parallel training scheme generates multiple forecasts from a single sample, which is computationally efficient. However, its success critically depends on perfectly causal normalization. The paper mentions this requirement but provides no details on its implementation or ablation studies demonstrating its effectiveness. If the normalization is not strictly causal, it could leak future information into predictions for shorter contexts, artificially inflating performance. A more detailed explanation and empirical validation are needed.

Incremental Contribution Relative to Existing Work: The SSM encoder is heavily based on S5, and the conceptual foundation of the decoder draws directly from HiPPO's insight of using hidden states to represent basis coefficients. The primary novelty lies in combining these ideas for forecasting rather than reconstruction. While this combination is effective, the individual components are not novel. The paper would benefit from a clearer delineation of which parts are genuinely new contributions versus engineering of existing methods.

---

> ### Author Rebuttal · Authors · 2026-03-31
>
> We thank the Reviewer for the insightful comments and feedback.
>
> **W1/Q1**: The Reviewer is correct that the selection of the scaling factor is currently based on a heuristic. However, this heuristic is straightforward and currently just relies on the sampling frequency and domain of the dataset, see Algorithm 2. In order to achieve the current state-of-the-art with FlowState, no additional tuning of the scaling factor for any dataset was necessary. Nevertheless, we fully agree that automatic selection of the scaling factor would be even better. Such seasonality-selection modules are largely independent of the pretrained forecasting model and can therefore be easily combined with FlowState; this approach has already been shown to work effectively in prior work, for example in TabPFN‑v2 [1].
>
> To assess this, we conducted a preliminary experiment with an automatic scale-factor selection variant, for which the results can be found in https://anonymous.4open.science/r/FlowState-ICML-75BF/README.md#Ablations. While it performs worse than regular FlowState, the degradation is moderate and concentrated to only a few datasets: **78** out of **97** tasks remain within 5% of the baseline, and many are identical to or even better than it. The main negative outliers arise in cases such as Solar Weekly, where the available context length is too short to capture even a single full season. Overall, these results suggest that with enough context, FlowState can be extended to work well without explicit knowledge of the sampling rate of the data.
>
> **W2/Q2**: We agree with the Reviewer that the coarser sampling of the data certainly will increase the equivariance error. However, we would like to point out that this is not an issue of the equivariance of FlowState, but rather a fundamental issue of data sampling itself. We further investigated the equivariance error, which confirmed our initial bound. A more detailed description can be found under **W2/W3** in the rebuttal to Reviewer **pABL**.
>
> **W3/Q3**: We agree that strictly causal normalization is critical for preventing information leakage in the parallel forecast training scheme. We would like to clarify that precisely for this reason we use Causal RevIN. It is formally defined in the main body of the paper (Eqs. 14–16), where the normalization statistics are explicitly constructed using running mean and standard deviation. We acknowledge that this may not have been sufficiently emphasized, and we will highlight this formulation more clearly in the revised manuscript.
>
> To further validate the necessity of causal normalization, we additionally conducted an ablation in which Causal RevIN is replaced with standard (non‑causal) RevIN, keeping the architecture, training procedure, and context length identical to FlowState‑3M (2k context). This change results in a consistent and noticeable performance degradation (MASE: from **0.725** to **0.738**), confirming that causal normalization is essential for reliable parallel training. In particular, standard RevIN introduces subtle leakage from future timesteps into predictions during training, which degrades generalization. These results can be found in the extended ablations, which we will include in the camera-ready version: https://anonymous.4open.science/r/FlowState-ICML-75BF/README.md#Ablations.
>
> These results empirically support the design choice of Causal RevIN. We will include this ablation and the corresponding results in the appendix of the camera‑ready version.
>
> **W4:Q4**: We agree with the Reviewer that the FBD is inspired by HiPPO, but to the best of our knowledge its application for decoding is novel. It uses polynomial basis functions, with the hidden states representing their coefficients, to produce the forecasts. Since our basis functions are orthogonal (or can even be orthonormal), they are proven to represent any function with arbitrary precision if enough polynomials are used. Therefore, it is sufficient that the FBD produces the proper coefficients for the polynomials and it is not required to also learn the polynomials themselves during training. To further illustrate this aspect, we conducted an experiment where we limited the number of polynomials used and observed performance regression, see “First 128 basis functions” and “First 64 basis functions” in https://anonymous.4open.science/r/FlowState-ICML-75BF/README.md#Ablations.
> Apart from the FBD another important novelty of FlowState is its training scheme employing parallel predictions.
>
> A common concern of the Reviewers was insufficient explanation of limitations. To address this, in the camera-ready version we will include the following dedicated section which can be seen under **Limitations** in the rebuttal for Reviewer **rokY**.
>
> [1] Hoo, Shi Bin, et al. "From Tables to Time: Extending TabPFN-v2 to Time Series Forecasting." arXiv preprint arXiv:2501.02945 (2025).

---

> > ### Author Rebuttal · Reviewer_wHTS · 2026-04-06
> >
> > Thank you for the thorough and constructive rebuttal. I appreciate the additional experiments and clarifications, particularly regarding automatic scaling factor selection, causal normalization, and the equivariance error discussion.
> >
> > The new ablation results on automatic scaling and causal RevIN are helpful and partially address my concerns about heuristic dependence and potential information leakage. The clarification on the decoder design and its relation to HiPPO is also useful in better understanding the contribution.
> >
> > The rebuttal strengthens the empirical grounding of the work, and I will take these updates into account in my final evaluation.

---

> > > ### Author Response · Authors · 2026-04-07
> > >
> > > We thank the reviewer for their thoughtful assessment. We are glad that the additional experiments and clarifications helped address the concerns regarding scaling factor selection, causal normalization, and equivariance error, and we appreciate the consideration in the final evaluation.

---

### Official Review · Reviewer_BZfQ · 2026-03-08

**Soundness:** 3
**Presentation:** 3
**Significance:** 4
**Originality:** 3
**Overall Recommendation:** 4
**Confidence:** 4

**Summary:**

This paper proposes a novel SSM time series foundation model architecture (TSFM). The model demonstrates several desirable properties such as (approximate) sampling rate equivariance, arbitrary length forecast generation, parameter efficiency relative to existing SOTA TSFMs. The authors use S5 style SSM blocks in the encoder with learnable step size in the ZOH discretization similar to other architectures featuring an SSM based encoder. One of the novel contributions is that the authors interpret the final SSM latent state token as coefficients for a fixed choice of continuous-time basis functions allowing them to sample forecasts with arbitrary sampling rate. Moreover, the authors empirically demonstrate sampling rate equivariance through a sampling rate scaling mechanism that allows the forecast to be sampled at a different sampling rate than the context.

Overall, the authors seem to push the pareto front in terms of forecasting performance and parameter efficiency on GIFT-eval.

**Compliance With Llm Reviewing Policy:**

Affirmed.

**Final Justification:**

The authors completely addressed my questions in the rebuttal and reinforced my initial assessment of the work as an exciting direction for performant, parameter-efficient TSFMs.

**Key Questions For Authors:**

#### Questions

1. Why did you not consider Mamba or any linear attention variants as the main sequence mixing mechanisms?
2. What are the failure modes of the model? Since the forecasts come from a choice of basis functions, does FlowState suffer from typical long horizon failure modes such as mean regression, divergence, etc?
3. (Related) How does FlowState handle very long context/horizon forecasting? This was a reported strength in the S4 paper.
4. Why is there no ablation for the time noise augmentation described in appendix C.1?

**Limitations:**

yes

**Strengths And Weaknesses:**

#### Strengths

1. The paper is well-written and clear to follow
2. Empirical measurements actually support the hand-wavy theoretical argument that the model is sampling rate equivariant
3. The parallel forecasting trick is an interesting pretraining feature enabled by SSMs that merits more follow-up work on it's own
4. The paper seems to be the first successful demonstration of SSM based TSFMs (on GIFT-eval)

#### Weaknesses

1. The SSM module is slightly outdated compared to the modern SSM variants such as Mamba which have readily available hardware efficient implementations e.g. in flash-linear-attention.
2. The temporal scaling strategy is fit to a particular forecasting regime and doesnt seem like it would transfer well to a more granular setting e.g. high-frequency financial time series.
3. Why is the evaluation limited to GIFT-eval? There are several time series benchmarks out there and it would be nice to get a sense on what domains FlowState is strong/weak on in particular.
4. (Wont affect score) There is a lack of qualitative forecast plots. It would be nice to see more forecasts at different granularities on more context windows for short, medium, and long term horizons in the appendix

---

> ### Author Rebuttal · Authors · 2026-03-31
>
> We thank the Reviewer for the insightful comments and feedback.
>
>
>
> **W1/Q1**: We acknowledge that S5 may seem slightly outdated and that considering more recent SSM variants could be beneficial, yet we opted for S5 as some of its aspects remain crucial. In particular, we investigated Mamba and Mamba2, but found their performance to be substantially worse than that of S5. This motivated us to perform two ablations “S5-real” and “selective $\Delta_E$” in Table 2, which evaluated two main differences between these architectures: A real state-transition matrix (or scalar) and an input-dependent $\Delta$ parameter. We found that both aspects regress performance and we hypothesize that for long forecasts, especially with varying context lengths and sampling rates, complex state transitions of S5 are essential. Apparently, the complex-valued recurrency can implement rotations, see Mamba-3[1] that might help to adjust the offset to the current position in the seasonal cycle, which is crucial for forecasting with varying context length. Selective delta also hurts performance, likely because it negatively impacts the adjustment of $\Delta$ in FlowState. However, although Mamba or Mamba-2 didn’t turn out to be a good replacement for S5, Mamba-3 brings back the complex-valued state-transition and might definitely be worthwhile investigating.
>
>
>
> **W2**: We deliberately selected the GIFT-Eval benchmark for evaluation, because it covers a wide range of forecasting regimes. It comprises datasets from various domains and with a diverse set of sampling frequencies, from yearly tourism and economic trends, down to several second sampling in Solar and Wind Power datasets. However, we did not evaluate FlowState on the specific case of high-frequency financial datasets. We would be interested in trying on those datasets, if the Reviewer can provide us suggestions.
>
>
>
> To quantitatively evaluate limitations with regard to various sampling regimes, we conducted a sensitivity analysis on the scale factor on the commonly used dataset ETTm1, see https://anonymous.4open.science/r/FlowState-ICML-75BF/README.md#sensitivity. We observed that the appropriate scaling factor of 0.25 works best and performance degrades for not-aligned scale factors.
>
>
>
> **Q2/Q3**: We investigated long context/horizon forecasting more in detail by providing a sine wave with varying context and target lengths to FlowState. As can be seen in https://anonymous.4open.science/r/FlowState-ICML-75BF/README.md#failure, we found no particular failure mode for longer contexts. In particular, the MSE error decreases until sufficient context is provided and then remained flat. In case of a varying target lengths, we can observe that FlowState achieves low MSE values for any reasonable forecasting length relevant for GIFT-Eval (up to 720)  which we focused on during pretraining, but indeed from thereon we see divergence and the MSE increases.
>
>
>
> **Q4**: We fully agree with the Reviewer that isolating the effect of time noise would further clarify the contribution of different components in FlowState. To address this, we conducted an additional ablation in which time noise is explicitly removed while keeping all other components unchanged. This results in performance degradation, highlighting the usefulness of time noise for robust scale‑equivariant modeling. The updated ablation results are shown here: https://anonymous.4open.science/r/FlowState-ICML-75BF/README.md#ablations.
>
> A common concern of the Reviewers was insufficient explanation of limitations. To address this, in the camera-ready version we will include the following dedicated section which can be seen under **Limitations** in the rebuttal for Reviewer **rokY**.
>
> [1] Lahoti, Aakash, et al. "Mamba-3: Improved Sequence Modeling using State Space Principles." 8 Oct. 2025, openreview.net/forum?id=HwCvaJOiCj.

---

> > ### Author Rebuttal · Reviewer_BZfQ · 2026-04-01
> >
> > I maintain my score.

---

> > > ### Author Response · Authors · 2026-04-07
> > >
> > > We thank the reviewer for their careful evaluation and are glad that their concerns have been adequately addressed.

---

### Official Review · Reviewer_pABL · 2026-03-11

**Soundness:** 3
**Presentation:** 2
**Significance:** 3
**Originality:** 3
**Overall Recommendation:** 5
**Confidence:** 3

**Summary:**

This paper addresses the problem of time scale, i.e. can a time series model work when the same underlying process is observed every 15 minutes, hour or weekly, where it also operates for different context and forecast lengths. The paper does so by proposing an S5-based state space encoder which operates on a continuous time design used with a functional basis decoder, which allows for decoding with different sampling rates and for different time horizons effectively. The paper also does pretraining, and shows very promising results on GIFT-Eval in comparison to other TSFMs.

**Compliance With Llm Reviewing Policy:**

Affirmed.

**Final Justification:**

Authors responded with a convincing and extensive rebuttal, which addressed most of my concerns. So, I increased my score accordingly to an "Accept" from "Weak Accept".

**Key Questions For Authors:**

1. It seems that the  scale factor is set using dataset level seasonality heuristic, which is outlined in Algorithm 2. Why does Transport, Healthcare,.., assigned automatically to have weekly frequency? Furthermore, could you elaborate on the sensitivity of results for different seasonalities chosen by the user?

2. The parameter counts are given but could you also provide wall-clock or flops type of measurement outlining the efficiency of FlowState? I think for efficiency claims, we also need to measure the computational complexity of the method.

I am inclined to raise my score once my concerns are addressed.

**Limitations:**

The impact of the paper has been discussed but there is no explicit discussion of the limitations.

**Strengths And Weaknesses:**

### Strengths

1. The motivation of this work is strong. Fixed models that are tied to one temporal resolution is a limitation and this work addresses specifically this. Figure 2 demonstrates that FlowState capably does that.
2. The Table 1 results are promising and shows that with smaller number of parameters, the model surpasses or almost matches far bigger models. The parameter efficiency of FlowState is significant.
3. In my opinion, functional basis decoder is quite novel (i.e., sampling scheme developed which first samples the continuous function and then decodes the outputs at given frequency).

 ### Weaknesses

1. It seems that the ablations in Table 2 do not isolate the functional basis decoder from an existential way. They vary the basis family, yet there is no encoder matched baseline where the same SSM backbone uses a standard fixed horizon decoder.  This ablation would be critical in understanding whether the performance improvement mainly stems from the functional basis decoder or the encoder itself.
2. In my opinion, the writing should be significantly improved. The exposure of the paper and the writing hinders understanding its contributions.
3. The theory section seems to be more about giving intuition than proof. In the current form, the equivariance arguments still make sense but without problem formulation/notation etc., it took a quite amount of time from my end to read and understand. Authors should extend their theory section either in the main section or the appendix.

---

> ### Author Rebuttal · Authors · 2026-03-31
>
> We thank the reviewer for the insightful comments and feedback.
>
>
>
> **W1**: We agree that isolating the contribution of the Functional Basis Decoder is important for understanding the source of FlowState’s performance gains. We conducted an additional ablation, in which we replaced the FBD with a resolution‑agnostic linear decoder operating on the same invariant SSM encoder representations, thereby removing the functional decoding mechanism while keeping the encoder fixed.
>
> This leads to a MASE degradation from 0.725 to 0.754, confirming that the FBD provides important contribution to the overall architecture. We hypothesize that the performance regression comes from the fact that with such a FBD the scale equivariance is not preserved, which leads to local forecasting errors. However, since our output generation scheme leverages Multi-Patch Inferencing, the predictions of the FBD are constrained to relatively small output patches, and the SSM encoder, together with Multi-Patch Inferencing, partially compensates the prediction on the global scale.
>
> Exemplary forecasts are included in https://anonymous.4open.science/r/FlowState-ICML-75BF/README.md#Fixed-Decoder. It can be clearly seen that on the global scale, the forecasts with the “Fixed decoer” look reasonable, but the individual forecasting patches exhibit significant deviations. We will include this result as a part of the ablation study, and attach the plots in the Appendix.
>
> Finally, the ablation labeled “w/o equivariance” in the submitted paper provides complementary evidence. In that setting, the FBD without scale adjustment also reduces to a linear layer, however, unlike the new ablation above, it additionally fixes the SSM discretization parameter. This leads to a significantly larger performance degradation, further highlighting the importance of jointly adapting both decoder and encoder to achieve scale equivariance.
>
> **W2/W3**: We acknowledge that the writing of the manuscript can be improved. In order to address this, we will do a thorough proofread for the camera-ready version, add a limitation section, see below, as well as clarify our notation in Section 4.
>
> The equivariance proof presented in this paper wasn’t considered as a strict mathematical proof, but we rather intended to provide intuitions, backed by empirical evidence through our experiments.
>
> Since then, we have more closely worked on a formal proof of the upper bounds and found that our initial proof intuition holds. Arriving at the following limit for an N-layer model with ZOH discretization:
>
> $$
> e_{\\mathrm{equi}}
> \\le
> \\|W_{\\mathrm{lin}}\\| \\, \\|P\\|_\\infty \\, N \\, \\frac{c \\, \\|C\\| \\, \\|B\\| \\, M (1+s)\\Delta}{2}
> (1 - e^{-\\alpha T}).
> $$
>
> where $\\|P\\|_\\infty$ denotes the basis function envelope, $c$ is a constant including the Lipschitz constant from the input signal and the constants from a stability condition on the eigenvalues of A, and $s = \Delta'/\Delta$ is the scale ratio. The error is $O(N(1+s)\Delta)$.
>
> We are happy to provide the complete formal proof if requested.
>
> **Q1**: For the domains “Transport”, “Healthcare”, “Sales”, etc. we implicitly assume that there are weekly cycles, because those domains are typically strongly correlated with the human work week. Apart from that, we did not handpick the scale factor for any particular dataset to achieve SOTA performance. The actual seasonality is then computed in the consecutive lines of Algorithm 2. We furthermore explored the sensitivity of FlowState to the choice of the scaling factor in an additional experiment shown in https://anonymous.4open.science/r/FlowState-ICML-75BF/README.md#sensitivity, which we will include in the appendix of the camera-ready version.
>
> **Q2**: The main focus of our manuscript was on the architectural development of a novel timeseries foundation model with capabilities beyond existing state-of-the-art models, which we demonstrated. However, we agree with the reviewer that adding inference-related metrics, such as FLOP counts is highly relevant. We conducted a preliminary experiment and observed that FlowState with a context of 2112 timesteps and a context length of 720 consumes \~124 GMACs, which is higher than LightGTS (\~220 MMACs), but much lower than TimesFM (\~625 GMACs) and other SOTA foundation models, see [1]. The main reason for the relatively higher number of MACs versus the parameter count of FlowState is that it doesn't employ input patching, which significantly shortens the time series sequence length to be processed.
>
> A common concern of the reviewers was insufficient explanation of limitations. To address this, in the camera-ready version we will include the following dedicated section which can be seen under **Limitations** in the rebuttal for Reviewer **rokY**.

---

> > ### Author Rebuttal · Reviewer_pABL · 2026-04-02
> >
> > I thank the authors for their detailed rebuttal and I also would like to thank for the experiments. Please add the complete proof to the paper/appendix. As my concerns were mostly addresses, I will increase my score accordingly.

---

> > > ### Author Response · Authors · 2026-04-07
> > >
> > > We thank the reviewer for the positive feedback and are glad that most concerns have been addressed. We will gladly include the formal proof in the appendix of the camera‑ready version.

---

### Official Review · Reviewer_rokY · 2026-03-12

**Soundness:** 3
**Presentation:** 3
**Significance:** 3
**Originality:** 3
**Overall Recommendation:** 5
**Confidence:** 4

**Summary:**

This paper proposes FlowState, a sampling-rate equivariant time-series foundation model. Built upon a core architecture that combines an SSM-based encoder with an innovative Functional Basis Decoder (FBD), FlowState enables continuous-time modeling and dynamic timescale adjustment. This design allows the model to naturally adapt to arbitrary sampling rates and flexibly adjust prediction horizons.

**Compliance With Llm Reviewing Policy:**

Affirmed.

**Final Justification:**

The rebuttal addressed your main concerns,

**Key Questions For Authors:**

See Weaknesses.

**Limitations:**

See Weaknesses.

**Strengths And Weaknesses:**

Strengths:
- The paper addresses an interesting research topic: forecasting time series data across diverse domains with varying sampling rates.
- The overall writing is clear and coherent.
- Sufficient theoretical background is provided to underpin the proposed model.

Weaknesses:
-  The literature review is somewhat insufficient. For instance, existing Time-series Foundation Models mentioned in the paper do not address the issue of varying sampling rates, whereas LightGTS specifically tackles this challenge.

- Regarding Table 1, the FlowState model does not achieve SOTA performance on GIFT-Eval; its MASE metric is outperformed by TimesFM-2.5, and its CRPS metric falls short of TiRex. Furthermore, could the authors evaluate the model's performance on standard datasets such as ETT, Weather, Traffic, and Electricity?

- FlowState bypasses patching in favor of point-wise tokenization. However, individual time points as tokens carry very limited information. How do the authors address this information density issue?

---

> ### Author Rebuttal · Authors · 2026-03-31
>
> We thank the Reviewer for the insightful comments and feedback.
>
> W1: Thank you also for bringing the LightGTS model to our attention. We focused our literature research primarily on models evaluated on foundation model benchmarks, such as GIFT eval. LightGTS is indeed highly relevant, and we will reference it in the camera-ready version.
>
> W2: We agree that SOTA performance is crucial for impact and to improve over the concern about slightly weaker results, we evaluated additional hyperparameter settings to close that gap. In particular, we first changed the context window for FlowState from 2048 to 4096 timesteps, resulting in FlowState-3M(4k) and FlowState-10M(4k). Secondly, we increased the size of the MLP layer, resulting in FlowState-18.6M(4k). Both, FlowState-10M(4k) and FlowState-18.6M(4k), achieve **state-of-the-art MASE** performance on the GIFT-Eval leaderboard with a MASE of **0.704** and **0.701** respectively, outperforming TimesFM-2.5 and TiRex. The 18.6M model further achieves state-of-the-art performance in CRPS with **0.487**. We will update Table 1 in the camera-ready version with the Table from https://anonymous.4open.science/r/FlowState-ICML-75BF/README.md#GIFT-Eval.
>
> We also evaluated FlowState on standard datasets, such as ETTh1, ETTm1, etc., where FlowState and LightGTS interchangeably share leading performance. Under https://anonymous.4open.science/r/FlowState-ICML-75BF/README.md#datasets one can find the comparison table, which we will also add to the appendix of the camera-ready version.
>
> W3: We agree that point-wise embedding can be problematic for architectures such as Transformers, as demonstrated in prior work [1]. However, we haven’t observed this issue with FlowState, which we believe is due to its use of SSMs.
>
> Compared to transformers, SSMs maintain a hidden state that accumulates information over multiple neighboring timesteps, making them well-suited to processing raw, high-resolution temporal signals. Prior work such as S4 [2] and S5 [3] has demonstrated strong performance on similarly fine-grained inputs, including raw audio. On top of that, FlowState explicitly controls the discretization parameter via a scaling factor, which allows it to dynamically adjust how much any individual timestep impacts the hidden state.
>
> **Limitations**: A common concern of the Reviewers was insufficient explanation of limitations. To address this, in the camera-ready version we will include the following dedicated section on limitations:
>
>
>     Despite its state-of-the-art performance, FlowState has limitations. Firstly, to determine the appropriate scale factor for each dataset it leverages a simple heuristic, see Algorithm 2, which though worked remarkably well across a variety of datasets. Secondly, FlowState does not natively support multi-variate data or covariates. This was a deliberate simplifying design choice, allowing to be agnostic to the specific number of channels or covariates, and to process them independently in uni-variate mode. Finally, these limitations provide grounds for future work and improvements. In particular, we plan to include automatic detection of the scale factor by decomposing the input time series into trend and seasonality, or by using Fourier analysis. We will also explore information exchange mechanisms between multi-variate inputs and covariates, such as the recently introduced grouped attention over the channels [1].
>
> [1] Zeng, Ailing, et al. ‘Are Transformers Effective for Time Series Forecasting?’ Proceedings of the AAAI Conference on Artificial Intelligence, vol. 37, no. 9, June 2023, pp. 11121–11128, https://doi.org/10.1609/aaai.v37i9.26317.
> [2]  Gu, Albert, et al. ‘Efficiently Modeling Long Sequences with Structured State Spaces’. International Conference on Learning Representations, 2022, openreview.net/forum?id=uYLFoz1vlAC.
> [3]  Smith, Jimmy T. H., et al. ‘Simplified State Space Layers for Sequence Modeling’. The Eleventh International Conference on Learning Representations, 2023, openreview.net/forum?id=Ai8Hw3AXqks.

---

> > ### Author Rebuttal · Reviewer_rokY · 2026-04-02
> >
> > Thank you for the author's reply. My problem has been solved, and I will increase my score.

---

> > > ### Author Response · Authors · 2026-04-07
> > >
> > > We thank the reviewer for taking our rebuttal into consideration and are glad that the concerns have been resolved.

---

### Decision · Program_Chairs · 2026-04-30

**Decision:**

Accept (regular)

**Comment:**

This paper presents a new time-series foundation model built on a state space model encoder with a functional basis decoder. The work is technically interesting, and the empirical evaluation shows a meaningful advantage over strong state-of-the-art baselines. A notable strength of the proposed approach is its flexibility in adapting to arbitrary sampling rates and varying prediction horizons. The reviewers provided detailed assessments, and the key concerns raised during the review process were appropriately addressed in the rebuttal. Considering the positive reviewer evaluations and the satisfactory author response, I recommend acceptance.